# Apiaceae Medicinal Plants in China: A Review of Traditional Uses, Phytochemistry, Bolting and Flowering (BF), and BF Control Methods

**DOI:** 10.3390/molecules28114384

**Published:** 2023-05-27

**Authors:** Meiling Li, Min Li, Li Wang, Mengfei Li, Jianhe Wei

**Affiliations:** 1State Key Laboratory of Arid Land Crop Science, Gansu Agricultural University, Lanzhou 730070, China; mlli1996@163.com (M.L.); lm1527431535@163.com (M.L.); 2Genome Analysis Laboratory of the Ministry of Agriculture and Rural Affairs, Agricultural Genomics Institute at Shenzhen, Chinese Academy of Agricultural Sciences, Shenzhen Branch, Guangdong Laboratory of Lingnan Modern Agriculture, Shenzhen 518120, China; wangli03@caas.cn; 3Institute of Medicinal Plant Development, Chinese Academy of Medical Sciences and Peking Union Medical College, Beijing 100193, China

**Keywords:** Apiaceae medicinal plants, traditional use, phytochemistry, bolting and flowering, controlling approaches

## Abstract

Apiaceae plants have been widely used in traditional Chinese medicine (TCM) for the removing dampness, relieving superficies, and dispelling cold, etc. In order to exploit potential applications as well as improve the yield and quality of Apiaceae medicinal plants (AMPs), the traditional use, modern pharmacological use, phytochemistry, effect of bolting and flowering (BF), and approaches for controlling BF were summarized. Currently, about 228 AMPs have been recorded as TCMs, with 6 medicinal parts, 79 traditional uses, 62 modern pharmacological uses, and 5 main kinds of metabolites. Three different degrees (i.e., significantly affected, affected to some extent, and not significantly affected) could be classed based on the yield and quality. Although the BF of some plants (e.g., *Angelica sinensis*) could be effectively controlled by standard cultivation techniques, the mechanism of BF has not yet been systemically revealed. This review will provide useful references for the reasonable exploration and high-quality production of AMPs.

## 1. Introduction

Apiaceae (syn. Umbelliferae) is one of the largest angiosperm families. It includes 300 genera (3000 species) globally and 100 genera (614 species) in China [1]. Apiaceae plants have been widely used in healthcare, nutrition, the food industry, and other fields [2]. Currently, 55 genera (230 species) of Apiaceae plants have been used as medicinal plants, and over 20 species have been widely used as traditional Chinese medicines (TCMs) [3]. Extensive studies have demonstrated that Apiaceae medicinal plants (AMPs) present a variety of pharmacological properties for the treatment of central nervous system, cardiovascular, and respiratory system diseases, amongst others [1,4]. These pharmacological activities are largely associated with metabolites such as polysaccharides, alkaloids, phenylpropanoids (simple phenylpropanoids and coumarins), flavonoids, and polyene alkynes [1,5,6].

In China, Apiaceae plants have been primarily used as traditional medicines for relaxing tendons, activating blood, relieving superficial wounds, treating colds, etc. [1,2]. For example, rhizomatous and whole plants are mainly used for the treatment of common colds, coughs, asthma, rheumatic arthralgia, ulcers, and pyogenes infections; fruits are mainly used for regulating vital energy, promoting digestion, relieving abdominal pain, and treating parasites [1,2].

The occurrence of bolting and flowering (BF) plays a critical role in the transition from vegetative growth to reproductive development in the plant life cycle [7]. However, BF significantly reduces the accumulation of metabolites in vegetative organs, which ultimately leads to the lignification of rhizomes and/or roots such as sugar beet [8], lettuce [9], and Chinese cabbage [10]. In particular, it common that BF significantly reduces the yield and quality of the rhizomatous AMPs [11]. Extensive studies have demonstrated that BF is regulated by both internal factors (e.g., germplasm resource, seedling size, and plant age) and external factors (e.g., vernalization, photoperiodism, and environmental stresses) [12]. To date, the BF of most rhizomatous AMPs have not been effectively controlled [11,13]. 

In order to form a comprehensive understanding of the current status of AMPs in China, herein, the progress on traditional use, phytochemistry, BF, and controlling approaches are summarized. This review will provide useful references for the efficient cultivation and quality improvement of AMPs.

## 2. Materials and Methods

Information on AMPs was attained using scientific databases (i.e., PubMed, Web of Science, Springer, and CNKI), using the following keywords: Apiaceae plant, traditional use, phytochemistry, BF, and lignification. Additional information was collected from ethnobotanical studies that mainly focused on the “*Flora of China*” and local classical literature, such as “*Divine Husbandman’s Classic of the Materia Medica* (*Shen Nong Ben Cao Jing*)”, “*Compendium of Materia Medica*”, “*Illustrated Book on Plants*”, “*Collection of National Chinese Herbal Medicine*”, and “*Pharmacopoeia of the People’s Republic of China”* (2020). The names of all plants correspond to the database *Catalogue of Life China*. Chemical structures were drawn using ChemDraw 21.0.0 software.

## 3. Apiaceae Medicinal Plants (AMPs)

Apiaceae plants have been traditionally used as medicines in China for ca. 2400 years (Figure 1). In 390–278 BC, three Apiaceae plants, including *Angelica dahurica*, *Ligusticum chuanxiong*, and *Cnidium monnieri,* were first recorded as medicines in “*Sorrow after Departure*” [1,2]. With the progress of Chinese civilization, ca. 100 Apiaceae plants were historically recorded as medicines. Specifically, 12 AMPs (e.g., *Angelica decursiva*, *Bupleurum chinense*, and *Centella asiatica*) were recorded in the known herbal text of China, the “*Divine Husbandman’s Classic of the Materia Medica* (*Shen Nong Ben Cao Jing*)” in 1st and 2nd century AD [14]. In 1578 and 1848, 24 and 31 AMPs were respectively recorded in the “*Compendium of Materia Medica* and *Illustrated Book on Plants”* [15]. In the 21st century, the number of AMPs has been continually increasing, up to 93 species recorded in the “*Flora of China*” in 2002 [16], and 96 species in the “*Collection of National Chinese Herbal Medicine*” in 2014 [17]. In recent years, 22 species were recorded in the “*Pharmacopoeia of the People’s Republic of China*” [18]. Specifically, 18 species are used with rhizomes and/or roots (Table 1).

## 4. Classification of AMPs Species

To our best knowledge, a total of 228 AMPs used as TCMs were collected from previously published studies and books (Table 1). Based on the traditionally used medicinal parts, the 228 AMPs were categorized into six classes, including 51 species (21 genera) used with the whole plants (i.e., rhizome and/or root, stem, and leaf), 184 species (44 genera) used with rhizomes and/or roots, 5 species (5 genera) used with stems, 9 species (8 genera) used with leaves, 17 species (14 genera) used with fruits, and 1 species (single genus) used with seeds. 

Specifically, the 51 species (21 genera) used with the whole plants include *Anethum*, *Anthriscus*, *Apium*, *Bupleurum*, *Centella*, *Conium*, *Coriandrum*, *Cryptotaenia*, *Eryngium*, *Ferula*, *Foeniculum*, *Hydrocotyle*, *Oenanthe*, *Peucedanum*, *Pimpinella*, *Pleurospermum*, *Pternopetalum*, *Sanicula*, *Sium*, *Spuriopimpinella*, and *Torilis* genera. In particular, *Sanicula* (e.g., *S. astrantiifolia*, *S. caerulescens*, *S. chinensis*), *Hydrocotyle* (e.g., *H. himalaica*, *H. hookeri*, and *H. nepalensis*), and *Pimpinella* (e.g., *P. candolleana*, *P. coriacea*, and *P. diversifolia*) genera plants are usually used as whole plants.

The 184 species (44 genera) used with the rhizomes and/or roots, which make up the majority of AMPs, include *Angelica*, *Anthriscus*, *Apium*, *Archangelica*, *Bupleurum*, *Carum*, *Changium*, *Chuanminshen*, *Cicuta*, *Cnidium*, *Conioselinum*, *Daucus*, *Eriocycla*, *Ferula*, *Foeniculum*, *Glehnia*, *Heracleum*, *Hymenidium*, *Kitagawia*, *Levisticum*, *Libanotis*, *Ligusticopsis*, *Ligusticum*, *Meeboldia*, *Nothosmyrnium*, *Oenanthe*, *Osmorhiza*, *Ostericum*, *Peucedanum*, *Phlojodicarpus*, *Physospermopsis*, *Pimpinella*, *Pleurospermum*, *Pternopetalum*, *Sanicula*, *Saposhnikovia*, *Selinum*, *Semenovia*, *Seseli*, *Seselopsis*, *Spuriopimpinella*, *Tongoloa, Torilis*, and *Vicatia* genera. Specifically, *Angelica* (e.g., *A. biserrata*, *A. dahurica*, and *A. sinensis*), *Bupleurum* (e.g., *B. bicaule*, *B. chinense*, and *B. scorzonerifolium*), and *Ligusticum* (*L. chuanxiong*, *L. jeholense*, and *L*. *sinense*) genera plants are usually used as rhizomes and/or roots.

The 5 species (5 genera) used with the stems include *Aegopodium* (*A. alpestre*), *Coriandrum* (*C. sativum*), *Foeniculum* (*F. vulgare*), *Ligusticum* (*L. chuanxiong*), and *Oenanthe* (*O. javanica*); the 9 species (8 genera) used with the leaves include *Aegopodium* (*A*. *alpestre*), *Anethum* (*A*. *graveolens*), *Angelica* (*A*. *morii*), *Anthriscus* (*A. nemorosa* and *A. sylvestris*), *Carum* (*C*. *carvi*), *Daucus* (*D. carota*), *Foeniculum* (*F*. *vulgare*), and *Ligusticum* (*L. chuanxiong*); the 17 species (14 genera) used with the fruits include: *Ammi* (*A. majus*), *Carum* (*C. buriaticum* and *C. carvi*), *Cnidium* (*C*. *monnieri*), *Coriandrum* (*C*. *sativum*), *Cuminum* (*C*. *cyminum*), *Cyclorhiza* (*C*. *peucedanifolia*), *Daucus* (*D. carota* L. and *D. carota* var. Carota), *Pimpinella* (*P. anisum*), *Trachyspermum* (*T. ammi*), and *Visnaga* (*V. daucoides*) genera; the single genera used with the seeds is *Ferula* (*F. bungeana*) (Table 1).

## 5. Traditional Uses

As is shown in Table 1, distinct traditional uses of the 228 AMPs were recorded. Based on their clinical agents, a total of 79 traditional uses are enriched, with 40 species contributing to the treatment of relieving pain, 36 species to the treatment of dispelling wind; and 21 species to the treatment of eliminating dampness (Figure 2).

Moreover, the AMPs were also widely used as “ethnodrugs” for ethnic minorities in China. For example, *Carum carvi* was used as Tibetan medicine for the treatment of dispelling wind and eliminating dampness, as well as treating cat fever and joint pain [86]; *Trachyspermum ammi* [236] was used as Uygur medicine for the treatment of eliminating cold damp, dispelling coldness, and promoting digestion; *Angelica acutiloba* was used in Korean medicine for the treatment of strengthening the spleen, enriching blood, stopping bleeding, and promoting coronary circulation [237]; *Angelica sinensis* was used as medicine for the Tujia minority for the treatment of enriching the blood, treating dysmenorrheal, and relaxing the bowel [238]; and *Chuanminshen violaceum* was used as a geo-authentic medicine of Sichuan province for the treatment of moistening the lungs, treating phlegm, and nourishing the spleen and stomach [89]. 

Meanwhile, AMPs combined with other herbs have also been applied for thousands of years [239]. For example, the Decoction of Notopterygium for Rheumatism is a famous Chinese prescription and is composed of *Notopterygium incisum*, *Angelica biserrata*, *Ligusticum sinense*, *Eryngium foetidum*, and *Ligusticum chuanxiong*, etc.; it has been widely used for the treatment of exopathogenic wind-cold, rheumatism, headache, and pantalgia [94]. The Xinyisan that is composed of *Yulania liliiflora*, *Actaea cimicifuga*, *Angelica dahurica*, *Eryngium foetidum*, *Ligusticum sinense*, etc., has been widely used for the treatment of deficiency of pulmonary qi and nasal obstruction due to wind-cold pathogens and damp-heat in the lung channel [94,168]. The Shiquan Dabu Wan of *Angelica sinensis* that is recorded in the “*Pharmacopoeia of the People’s Republic of China*” has been mainly used for the treatment of pallor, fatigability, and palpitations [240]. The Juanbi Tang of *Notopterygium incisum* and *Angelica biserrata* that is recorded in “*Medical Words*” (Qing dynasty) has been mainly used for treatment of arthralgia due to wind cold-dampness [121].

## 6. Modern Pharmacological Uses

Modern pharmacological research on the 228 AMPs is summarized in Table 1. Based on the pharmacological effects, a total of 62 modern uses are identified (Figure 3), with 36 species showing anti-inflammatory activity, 20 species showing antioxidant activity, and 16 species showing antitumor activity. In addition, other modern uses are also identified, such as antitumor, bacteriostatic, and analgesic. These modern pharmaceutical properties have been demonstrated to be associated with bioactive metabolites, and several metabolites have been found to be co-existent in the TCMs [241,242].

Specifically, sesquiterpene-coumarin, such as (3′S, 5′S, 8′R, 9′S, 10′R)-kellerin, gummosin, galbanic acid, and methyl galbanate from *Ferula sinkiangensis* resin, showed anti-neuroinflammatory effects and might be a potential natural therapeutic agent for Alzheimer’s disease [243]. The supercritical carbon dioxide extracts from *Apium graveolens* showed antibacterial effects, with the highest inhibitory activity against *Bacillus cereus* [244,245]. In vitro, the antitumor activity of AMPs have been identified; for example, the ferulin B and C in *Ferula ferulaeoides* rhizomes could restrain the multiplication of HepG2 stomach cancer cell lines, and 2,3-dihydro-7-hydroxyl-2R*, 3R*-dimethyl-2-[4,8-dimethyl-3(E),7-nonadienyl]-furo [3,2-c] coumarin could restrain the proliferation of HepG2, MCF-7, and C6 cancer cell lines [107,246]. In addition, the osthole in *Angelica biserrata* could restrain the multiplication of human gastric cancer cell lines MKN-45 and BGC-823, human lung adenocarcinoma cell line A549, human mammary carcinoma cell line MCF-7, and human colon carcinoma cell line LOVO [247]. The antioxidative activity of AMPs has been also identified; for example, the imperatorin, oxypeucedanin hydrate, and bergaptol in *Angelica dahurica* exhibited DPPH scavenging activity [30], hydromethanolic extracts from *Pimpinella anisum* exhibited free radical scavenging activity [248], and water-soluble polysaccharides in *Chuanminshen violaceum* scavenged DPPH, hydroxyl, and superoxide anion radicals [91].

## 7. Phytochemistry

As is shown in Table 1, hundreds of bioactive metabolites have been identified from the 228 AMPs [1,249]. Based on their chemical structures, these metabolites can be categorized into five main classes: (1) polysaccharides, (2) alkaloids, (3) phenylpropanoids, (4) flavonoids, and (5) terpenoids (Figure 4).

Among the 22 AMPs recorded in the “*Pharmacopoeia of the People’s Republic of China*” [18], 18 secondary metabolites in the 17 AMPs (e.g., *Angelica biserrata*, *Bupleurum chinense DC.*, and *Centella asiatica*) (Figure 5) were described as quality control indicators, which include: 10 phenylpropanoids (i.e., osthole, columbianadin, imperatorin, isoimperatorin, nodakenin, ferulic acid, trans-anethole, notopterol, praeruptorin A, and praeruptorin B), 4 terpenoids (i.e., saikosaponin a, saikosaponin d, asiaticoside, and madecassoside), 2 chromones (i.e., prim-O-glucosylcimifugin and 5-O-methylvisammioside), and 2 phthalides (i.e., ligustilide and levistilide A); a specific quality marker has not been reported for the other 5 AMPs (e.g., *Changium smyrnioides*, *Daucus carota* L., and *Glehnia littoralis*) (Table 2).

### 7.1. Polysaccharides

Polysaccharides are the largest components of biomass and account for ca. 90% of the carbohydrates in plants [250]. Studies have demonstrated that polysaccharides in medicinal plants are indispensable bioactive compounds, presenting uniquely pharmacological effects such as immunomodulatory, hypoglycemic, antitumor, anti-diabetic, and antioxidant effects, amongst others, with few side effects or adverse drug reactions [251,252]. To date, polysaccharides in the 228 AMPs have also been identified, showing multiple pharmacological effects. For example, polysaccharides in *Angelica sinensis* present hematopoietic, antitumor, and liver protection effects [239,253]; polysaccharides in *Angelica dahurica* protect spleen lymphocytes, natural killer cells, and procoagulants [254,255]; and polysaccharides in *Bupleurum chinense* and *Bupleurum smithii* present the effect of macrophage modulation, kidney protection, and inflammatory alleviation [256,257,258].

### 7.2. Alkaloids

About 27,000 alkaloids presenting as water-soluble salts of organic acids, esters, and combined with tannins or sugars have been found in plants [259]. Many alkaloids are valuable medicinal agents that can be utilized to treat various diseases, including malaria, diabetes, cancer, cardiac dysfunction, blood clotting–related diseases, etc. [260,261,262]. Alkaloids in the 228 AMPs mainly exist in the *Ligusticum*, *Apium*, *Conium*, and *Cuminum* genera [249]. Pharmacological studies have demonstrated that alkaloids in *Ligusticum chuanxiong* show the activity of inhibiting myocardial fibrosis, protecting ischemic myocardium, and relieving cerebral ischemia-reperfusion injury [151,263,264]. A novel alkaloid 2-pentylpiperidine known as conmaculatin in *Conium maculatum* shows strong peripheral and central antinociceptive activity [265]. Some alkaloids have been identified to show antidepressant activity, such as berberine in *Berberis aristata*, strictosidine acid in *Psychotria myriantha*, and Anonaine in *Annona cherimolia*; these could be explored as an emerging therapeutic alternative for the treatment of depression.

### 7.3. Phenylpropanoids

Phenylpropanoids are a large class of secondary metabolites biosynthesized from amino acids, phenylalanine, and tyrosine [266]. Over 8000 aromatic metabolites of the phenylpropanoids have been identified in plants. These include simple phenylpropanoids (propenyl benzene, phenylpropionic acid, and phenylpropyl alcohol), coumarins, lignins, lignans, and flavonoids [267].

#### 7.3.1. Simple Phenylpropanoids

To date, limited simple phenylpropanoids have been identified from AMPs, including three phenylpropanoids (trans-isoelemicin, sarisan, and trans-isomyristicin) in the roots of *Ligusticum mutellina* [268]. Ferulic acid, one of the phenylpropionic acids, is an important bioactive metabolite of AMPs; it mainly exists in *Angelica*, *Ligusticum*, *Ferula*, and *Pleurospermum* genera [239,269,270]. Pharmacological studies have demonstrated that the ferulic acid in *Angelica sinensis* shows strong properties in inhibiting platelet aggregation, increasing coronary blood flow, and stimulating smooth muscle [271,272]; the ferulic acid in *Angelica acutiloba* shows antidiabetic, immunostimulant, antiinfammatory, antimicrobial, anti-arrhythmic, and antithrombotic activity [273]; and the ferulic acid in *Ligusticum tenuissimum* shows anti-melanogenic and anti-oxidative effects [274].

#### 7.3.2. Coumarins

Coumarins are the most widespread in 20 genera of AMPs (e.g., *Angelica*, *Bupleurum*, and *Peucedanum*) and mainly include simple coumarins, pyranocoumarins, and furocoumarins [56,275,276]. In recent years, distinct coumarins have been identified from AMPs, such as 99 coumarins in *Ferula* [277], 116 coumarins in *Angelica decursiva* and *Peucedanum praeruptorum* [180], and 9 coumarins in *Angelica dahurica* [278]. Furthermore, 8 coumarins were selected as quality markers, including osthole (1) in *Angelica biserrata* and *Cnidium monnieri*; columbianadin (2) in *Angelica biserrata*; imperatorin (3) in *Angelica dahurica* and *Angelica dahurica* cv. Hangbaizhi; isoimperatorin (4) in *Angelica dahurica*, *Angelica dahurica* cv. Hangbaizhi, *Notopterygium franchetii*, and *Notopterygium incisum*; nodakenin (5) in *Angelica decursiva*, *Notopterygium franchetii*, and *Notopterygium incisum*; notopterol (8) in *Notopterygium franchetii* and *Notopterygium incisum*; and praeruptorin A (9) and praeruptorin B (10) in *Peucedanum praeruptorum* (see Table 2 and Figure 5) [18].

To date, various biological activities of coumarins have been demonstrated, including antifungal, antimicrobial, antiviral, anti-cancerous, antitumor, anti-inflammatory, anti-filarial, enzyme inhibitory, antiaflatoxigenic, analgesic, antioxidant, and oestrogenic [279,280,281,282]. For example, coumarins are recognized as the main bioactive constituents in *Peucedani* genus and play critical roles in relieving cough and asthma, strengthening heart function, as well as preventing and treating cardiovascular diseases such as nodakenin, (+)-praeruptorin B, and praeruptorin C [283]; imperatorin oxypeucedanin hydrate, xanthotoxol, bergaptol, 5-methoxy-8-hydroxypsoralen, isoimperatorin, phelloptorin, and pabularinone in *Angelica dahurica* exhibit moderate DPPH scavenging activity, strong ABTS^·+^ scavenging activity, and significant inhibition on HepG2 cells, which could be explored as new and potential natural antioxidants and cancer prevention agents [30]; pabulenol and osthol extracts from *Angelica genuflexa* show anti-platelet and anti-coagulant components [38]; and decursinol angelate in *Angelica gigas* shows platelet aggregation and blood coagulation activity [38].

### 7.4. Flavonoids

Flavonoids are a group of the most abundant secondary metabolites in plants [266]. Generally, flavonoids can be further categorized into eight subgroups, including flavones (e.g., apigenin, luteolin, and baicalein), flavonols (e.g., kaempferol, quercetin, and myricetin), flavanones (e.g., naringenin, hesperitin, and liquiritigenin), flavanonols (e.g., dihydrokaempferol, dihydromyricetin, and dihydroquercetin), isoflavones (e.g., daidzein, purerarin, and peterocarpin), aurones, anthocyanidins, and proanthocyanidins [284,285,286]. In recent years, flavonoids have been identified from AMPs, such as 6 flavonoids (e.g., luteolin, isoquercitrin, and rutin) in *Ferula* [107], 12 flavonoids (e.g., quercetin-3-*O*-rutinoside, kaempferol-3,7-di-*O*-rhamnoside, quercetin-3-*O*-arabinoside) in Bupleurum [287], and 18 flavonoids (e.g., rutin, quercetin, and quercitrin) in Hydrocotyle [135].

To date, various biological activities of flavonoids have been demonstrated, including antioxidant, antiinflammatory, antidiabetic, anticancer, antiobesity, and cardioprotective [284,288]. For example, the apigenin in *Apium graveolens* shows anticancer properties [21], flavonoids in *Pimpinella diversifolia* DC.*, Anthriscus sylvestris*, and *Sanicula astrantiifolia* show antioxidant effects [197,289], and quercetin and its metabolites show vasodilator effects, with selectivity toward the resistance vessels [290].

### 7.5. Terpenoids

About 25,000 terpenoids have been reported in plants; they are diverse secondary metabolites containing three subgroups, including monoterpenoids, sesquiterpenes, and triterpenoids [291]. To date, terpenoids have been also identified in AMPs, such as 4 terpenoids (e.g., angelicoidenol, pregnenolone, and β-sitosterol) in *Pleurospermum* [142], 75 terpenoids (e.g., myrcene, farnesene, and xiongterpene) in *Ligusticum* [141], 109 terpenoids (e.g., nerolidol, guaiol, and ferulactone A) in *Ferula* [277], and 13 triterpenoids (e.g., ranuncoside, oleanane, and barrigenol) in *Hydrocotyle sibthorpioides* Lam. [136]. Specifically, saikosaponin triterpenes constitute the main class of secondary metabolites in the genus *Bupleurum*, with more than 90 saponins (e.g., saikosaponin a, b, and c) isolated [64,292].

Studies have found that terpenoids possess various biological activities, including anti-inflammatory, anti-oxidative, anti-fibrosis, antitumor, anti-Alzheimer’s disease, and anti-depression activities [293,294]. For example, the xiongterpene in *Ligusticum chuanxiong* shows insecticide effects [151], the asiaticoside in *Centella asiatica* shows antitumor properties [295], and the saikosaponin d in *Bupleurum chinense* DC. and *Bupleurum scorzonerifolium* show the effects of reducing blood glucose, inhibiting inflammation, and reducing insulin resistance [296].

### 7.6. Other Compounds

Chromones and phthalides also exist in AMPs and show pharmacological properties. Specifically, phthalides (e.g., ligustilide, *n*-butylidenephthalide, and *Z*-ligustilide) in *Angelica sinensis* show the effect of inhibiting vasodilation, decreasing platelet aggregation, as well as exerting analgesic, anti-inflammatory, and anti-proliferative effects [239]; butylphthalide in *Ligusticum sinense* shows anti-inflammatory and antithrombus effects, dilates blood vessels, and improves brain microcirculation and anti-myocardial ischemia [155].

In terms of chromones, 3 chromones (i.e., 5 thydroxy 2 [(angebyloxy) mehyI] fuan [3, 2′: 6, 7] chrmone, angeliticin A, and noreugenin) in *Angelica polymorpha* [297], 10 chromones (e.g., cnidimoside A, cnidimol B, and peucenin) in *Cnidiummonnieri* (L.) Cuss. [93], and 22 chromones (e.g., edebouriellol, hamaudol, and 3′(R)-(+)-hamaudol) in *Saposhnikovia divaricate* [218] have been identified. Studies have found that two chromones 3′S-(-)-*O*-acetylhamaudol and (±)-hamaudol in *Angelica morii* show the effect of inhibiting Ca^2+^ influx of vascular smooth muscle [298], prim-*O*-glucosylcimifugin and 5-*O*-methylvisammioside show antipyretic, analgesic, and anti-inflammatory effects [299], and chromones in *Bupleurum multinerve* show analgesic effects [300].

## 8. Effect of Bolting and Flowering (BF) on Yield and Quality

Previous studies have repeatedly emphasized that BF reduces the yield and quality of plants, especially in rhizomatous medicinal plants [11]. Here, a total of 38 rhizomatous plants that have been reported in the 228 AMPs are associated with BF (Table 3). Based on the effect degree of BF on the yield and quality, 38 rhizomatous AMPs belonging to 17 genera can be categorized into 3 classes: (1) BF significantly affects the yield and quality of 14 AMPs (i.e., *Angelica acutiloba*, *Angelica biserrata*, *Angelica dahurica*, *Angelica dahurica* cv. Hangbaizhi, *Angelica decursiva*, *Angelica polymorpha*, *Angelica sinensis*, *Daucus carota*, *Heracleum hemsleyanum*, *Heracleum rapula*, *Libanotis iliensis*, *Libanotis seseloides*, *Peucedanum praeruptorum*, and *Saposhnikovia divaricata*), and their rhizomes and/or roots are wholly lignified and cannot be used for clinical application; (2) BF affects the yield of 11 AMPs (i.e., *Angelica gigas*, *Bupleurum chinense*, *Bupleurum scorzonerifolium*, *Changium smyrnioides*, *Chuanminshen violaceum*, *Glehnia littoralis*, *Ligusticum chuanxiong*, *Ligusticum jeholense*, *Ligusticum sinense*, *Notopterygium franchetii*, and *Notopterygium incisum*), though their rhizomes or roots can be used as medicine to some extent; (3) BF has no significant effect on the yield and quality of 13 AMPs (i.e., *Angelica sylvestris*, *Cicuta virosa*, *Ferula ferulaeoides*, *Ferula fukanensis*, *Ferula lehmannii*, *Ferula olivacea*, *Ferula sinkiangensis*, *Ferula teterrima*, *Levisticum officinale*, *Libanotis buchtormensis*, *Libanotis lancifolia*, *Libanotis spodotrichoma*, and *Pimpinella candolleana*), and their rhizomes or roots can be used as medicine (Figure 6).

For example, for class (1) after BF, there was a 8.3- and 16.1-fold reduction of dry weight and quality marker ferulic acid content in *Angelica sinensis* [301] and a 1.5- and 1.5-fold reduction of dry weight and quality marker isoimperatorin content in *Angelica dahurica* [302]. For class (2), there was a 1.34-fold reduction of saikosaponinsands, while no significant change of dry weight in *Bupleurum chinense* was seen [303,304]; and a 2.0- and 1.7-fold reduction of dry weigh and polysaccharide content in *Changium smyrnioides* [305]. For class (3), there was no reduction of the yield and quality of the 13 AMPs at the harvest stages [19].

## 9. Approaches to Control BF

Generally, most Apiaceae plants are “low-temperature and long-day” perennial herbs; in other words, the plants must experience vernalization (i.e., an extended period of cool weather at 0 to 10 °C) and long days (>12 h daylight) to induce BF. Examples include *Angelica sinensis* [325], *Daucus carota* [326], and *Coriandrum sativum* [327].

Table 4 shows the approaches to inhibit BF of 24 AMPs. For example, the bolting rate of *Angelica sinensis* can be significantly decreased by planting the green stem cultivar (Mingui 2) instead of the purple stem cultivar (Mingui 1) [328], selecting smaller seedlings (i.e., root-shoulder diameter <0.55 cm) instead of larger seedlings [329,330], storing the seedlings at freezing temperature (i.e., <0 °C) during the overwinter stage [325], shading the plants under sunshade (i.e., >40%) during growth stage [331], and providing the plants with good growth conditions (e.g., plant intensity, nutrient and water balance) [332]. The bolting rate of *Angelica dahurica* can be significantly decreased through planting pure breeds [333], selecting immature seeds for seeding [308], increasing potassic fertilizer while decreasing nitrogen and phosphorus fertilizers [334], and planting using standard techniques [335]. The bolting rate of *Saposhnikovia divaricata* can also be significantly decreased by controlling the sunshade [336], sowing date [337], and planting density [338], and preventing excessive growth [336].

To inhibit the occurrence of BF in AMPs, several measures can be used, including breeding new cultivars, controlling the seedling age and size to delay the transition from vegetative growth to flowering, storing seedlings at freezing temperatures to avoid vernalization, growing the plants under sunshade to avoid long-day photoperiodism, and planting with standard techniques to reduce pests and diseases (Figure 7).

## 10. The Mechanism of BF Inducing the Rhizome Lignification

Extensive experiments have demonstrated that BF induces the lignification of fleshy rhizomes and enhances the degradation of metabolites [11,13,328]. Studies on anatomical structures reveal that the ratio of secondary phloem to secondary xylem respectively changes from 2:1 to 1:10 and 2/5–1/2 to 1/2–3/4 for the rhizomes of *Angelica sinensis* and *Angelica dahurica* before and after BF; meanwhile, the number of secretory cells producing essential oils significantly decreased [368,369]. Studies have found that the Early Bolting In Short Day (EBS) acts as a negative transcriptional regulator, preventing premature flowering of *Arabidopsis thaliana*, and co-enrichment of a subset of EBS-associated genes with H3K4me3, H3K27me3, and Polycomb repressor complex 2 has been observed [370]; a potential genetic resource for radish late-bolting breeding with introgression of the RsVRN1In-536 insertion allele into the early-bolting genotype could contribute to delayed bolting time of *Raphanus sativus* [371]; and *peroxidases* (*PRXs*) involved in lignin monomer biosynthesis were found to be down-regulated in *Peucedanum praeruptorum* at the bolting stage [372].

As is known, lignin biosynthesis belongs to the general phenylpropanoid pathway, which starts from phenylalanine and is catalyzed by a series of enzymes [13,373]. Specifically, phenylalanine is catalyzed to form *p*-Coumaroyl CoA sequentially through the three enzymes phenylalanine ammonia lyase (PAL), cinnamate 4-hydroxylase (C4H), and 4-coumarate-CoA ligase (4CL). Lignin biosynthesis is synthesized via three sub-pathways, including the following: (1) lignins are catalyzed to from *p*-Coumaroyl CoA sequentially through the three enzymes cinnamoyl-CoA reductases (CCR), cinnamyl alcohol dehydrogenases (CAD), and laccases (LACs), and then coniferyl aldehyde is catalyzed to from *p*-Coumaroyl CoA sequentially through the four enzymes hydroxycinnamoyl shikimate/quinate transferase (HCT), *p*-coumarate 3-hydroxylase (C3H), caffeoyl-CoA 3-*O*-methyltransferase (CCOMT), and CCR; (2) lignins are catalyzed to from coniferyl aldehyde sequentially through the two enzymes CAD and LAC; (3) lignins are catalyzed to from coniferyl aldehyde sequentially through the three enzymes ferulate 5-hydroxylase (F5H), caffeic acid 3-*O*-methyltransferase (COMT), and LACs (Figure 8).

Although lignin biosynthesis has been depicted, studies on the mechanism of BF inducing rhizome lignification are still limited. To date, the mechanism of BF affecting *Angelica sinensis* has been revealed, with the expression level of genes (e.g., *PAL1*, *4CLs*, *HCT*, *CAD1*, and *LACs*) significantly upregulated at the stem-node forming and elongating stage compared with the stem-node pre-differentiation stage, leading to the reduction of accumulation of secondary metabolites (i.e., ferulic acid and flavonoids) [13].

## 11. Conclusions and Future Aspects

In this review, we summarized the history of AMPs as TCMs, the classification of AMPs species, their traditional use, modern pharmacological use, and phytochemistry; the effect of BF on yield and quality, approaches to control BF, and the mechanisms of BF, inducing rhizome lignification. Although ca. 228 AMPs, 79 traditional uses, 62 modern uses, and 5 main kinds of metabolites have been recorded, the potential properties remain to be exploited. Although BF significantly reduces the yield and quality of AMPs, effective measures to inhibit BF have not been applied in the field, and the mechanisms of BF have not been systemically revealed for most AMPs. Thus, in order to effectively control the BF of AMPs to improve their quality and yield, on the one hand, standard cultivation techniques of AMPs should be applied; on the other hand, new cultivars should be developed by modern biotechnology such as the CRISPR/Cas9 system.

## Figures and Tables

**Figure 1 molecules-28-04384-f001:**
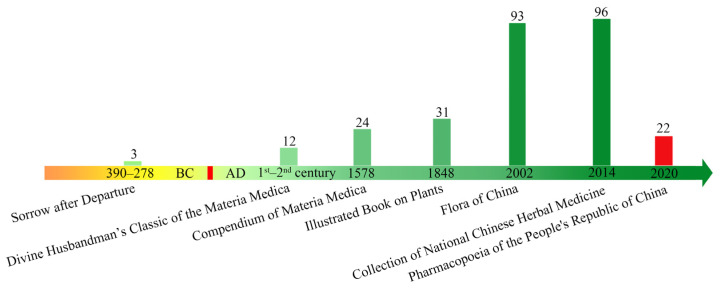
Apiaceae medicinal plants (AMPs).

**Figure 2 molecules-28-04384-f002:**
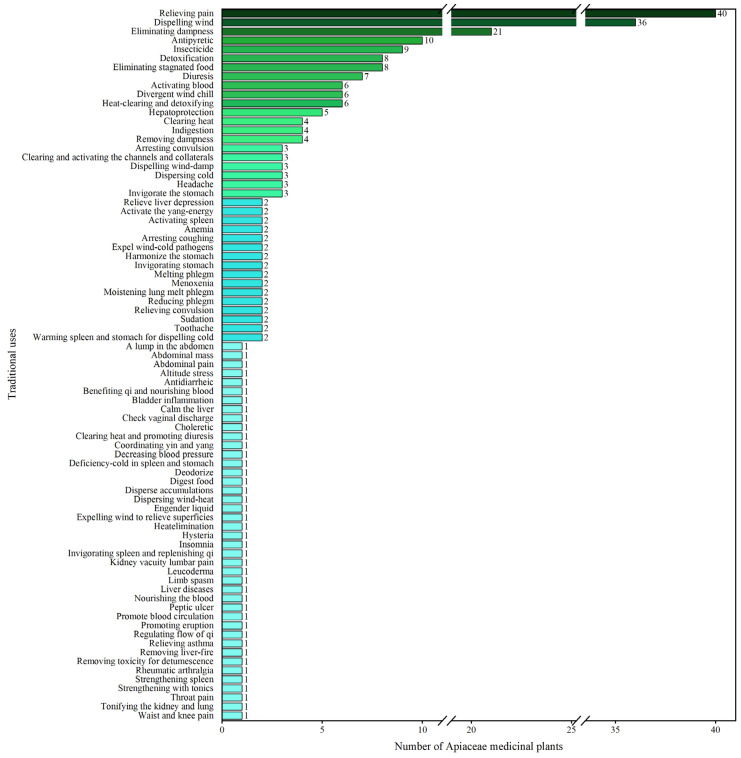
Traditional use of the 228 AMPs.

**Figure 3 molecules-28-04384-f003:**
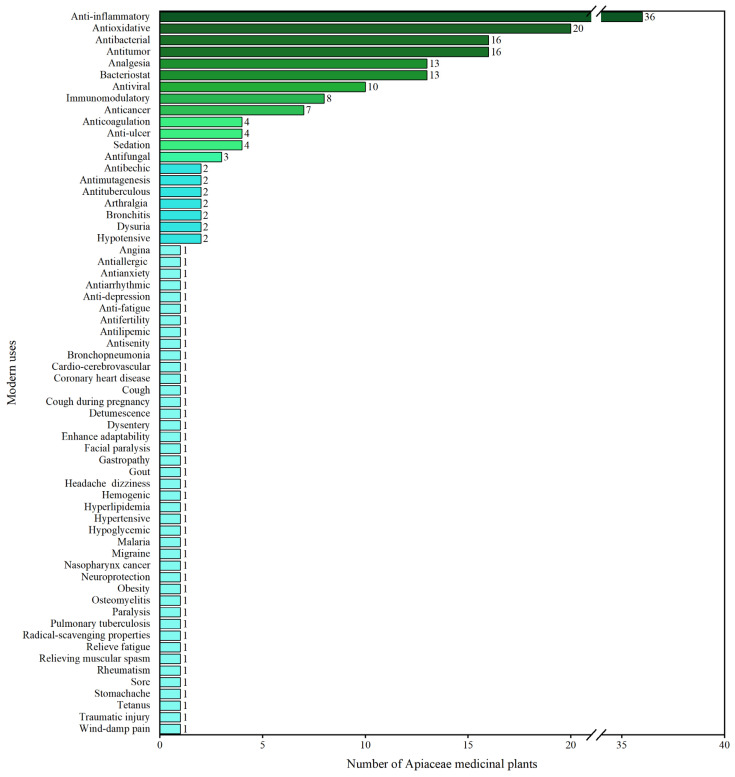
Modern pharmacological uses of the 228 AMPs.

**Figure 4 molecules-28-04384-f004:**
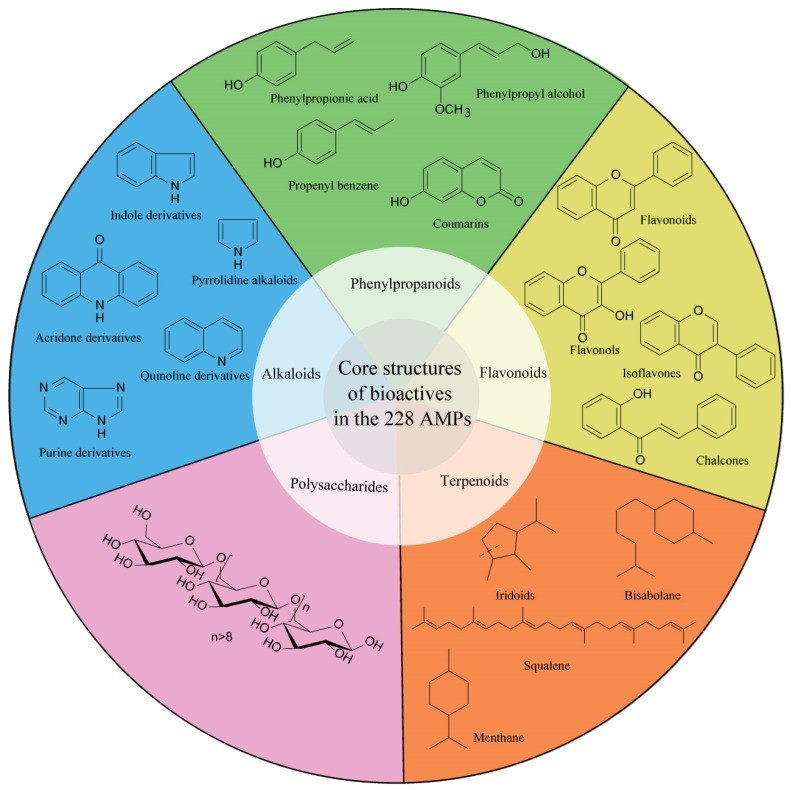
Core structures of five different bioactive compounds identified from the 228 AMPs.

**Figure 5 molecules-28-04384-f005:**
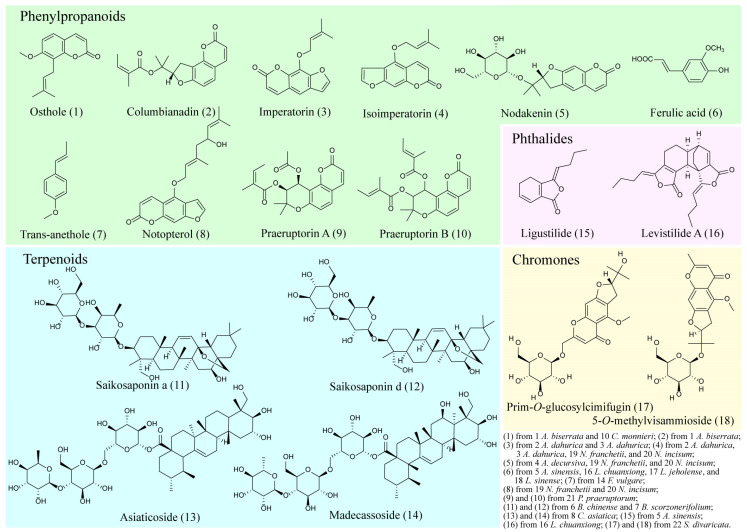
Structures of the 18 quality markers from the 22 AMPs in the “*Pharmacopoeia of the People’s Republic of China”* (2020) [18].

**Figure 6 molecules-28-04384-f006:**
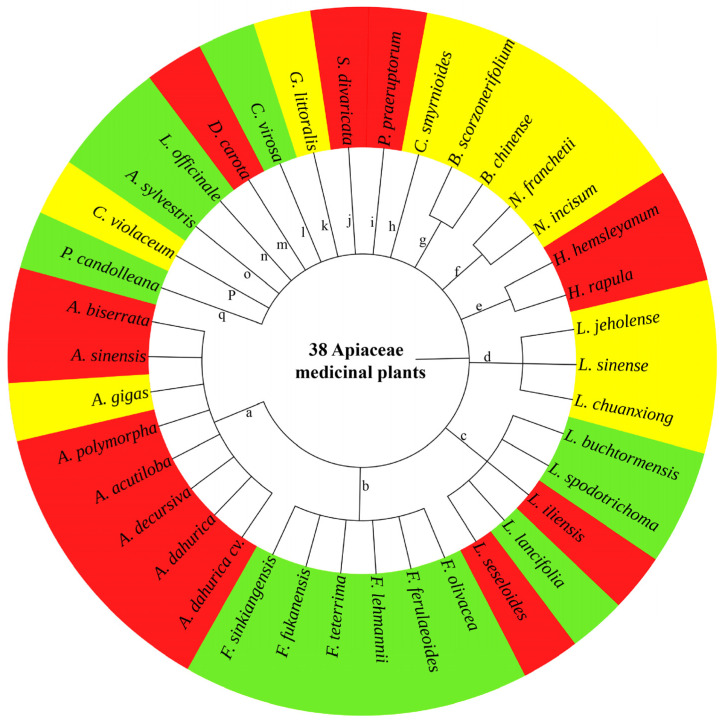
Cluster of the 38 rhizomatous AMPs affected by bolting and flowering (BF). The red color indicates that BF significantly affects the yield and quality; the yellow color indicates that BF affects the yield, though the rhizomes or roots can be used as medicine to some extent; and the green color indicates that BF has no significant effect on the yield and quality. a: *Angelica*, b: *Ferula*, c: *Libanotis*, d: *Ligusticum*, e: *Heracleum*, f: *Notopterygium*, g: *Bupleurum*, h: *Changium*, i: *Peucedanum*, j: *Saposhnikovia*, k: *Glehnia*, l: *Cicuta*, m: *Daucus*, n: *Levisticum*, o: *Anthriscus*, p: *Chuanminshen*, and q: *Pimpinella*.

**Figure 7 molecules-28-04384-f007:**
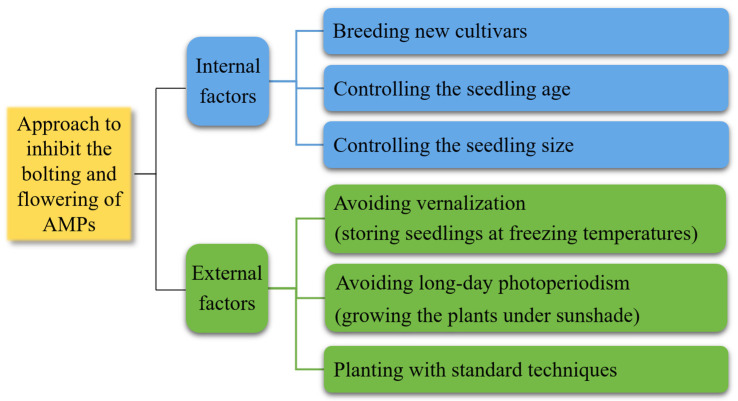
Approaches to control the BF of AMPs.

**Figure 8 molecules-28-04384-f008:**
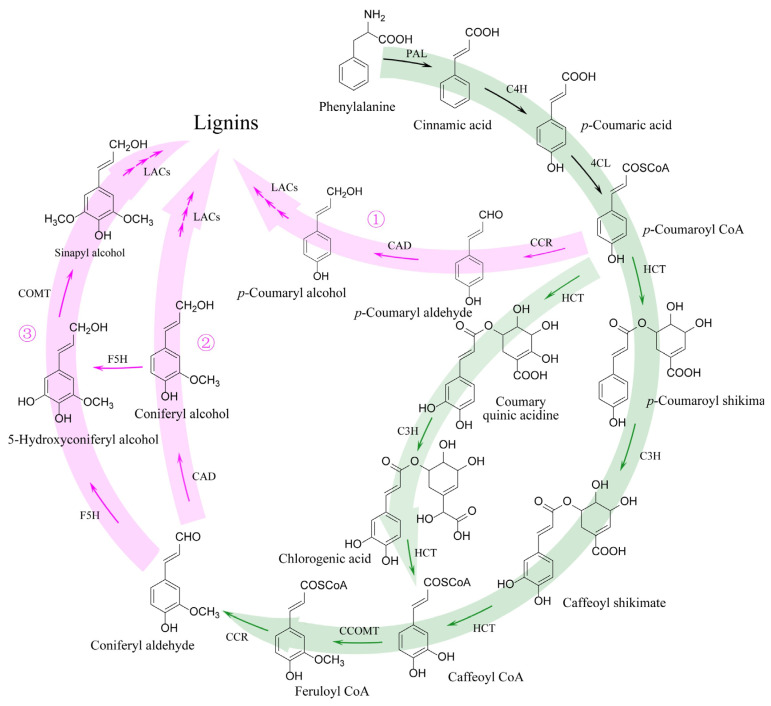
Schematic representation of biosynthetic pathways of lignins. Abbreviations: PAL, phenylalanine ammonia lyase; C4H, cinnamate 4-hydroxylase; 4CL, 4-coumarate-CoA ligase; HCT, hydroxycinnamoyl shikimate/quinate transferase; C3H, *p*-coumarate 3-hydroxylase; CCOMT, caffeoyl-CoA 3-*O*-methyltransferase; CCR, cinnamoyl-CoA reductases; CADs, cinnamyl alcohol dehydrogenases; LACs, laccases; F5H, ferulate 5-hydroxylase; COMT, caffeic acid 3-*O*-methyltransferase. The green color indicates the common phenylpropanoid pathway of phenylpropanoids, and the red color indicates the lignin biosynthetic sub-pathway. The ①, ② and ③ means different sub-pathways of lignin biosynthesis.

**Table 1 molecules-28-04384-t001:** The classification, traditional use, modern pharmacological use, and main metabolites of the 228 AMPs.

No.	Plant Species	Local Name in Chinese	Parts of Plant Used	Traditional Use	Modern Pharmacological Use	Main Metabolites	References
1	*Aegopodium alpestre* Ledeb.	Xiaoyeqin	Stems and leaves	Dispelling wind, relieving pain, and treating influenza	Treatment of rheumatic diseases, obesity, and hypotensive	Apiole, undecane, and limonene	[19,20,21]
2	*Ammi majus* L.	Daamiqin	Fruits	Treatment of vitiligo	\	Furanocoumarins	[16]
3	*Anethum graveolens* L.	Shiluo	Fruits, leaves, or whole plant	Treatment of bladder inflammation, liver diseases, and insomnia	Antibacterial, antifungal, antioxidant	Alkaloid, terpenoids, and flavonoids	[22]
4	*Angelica acutiloba* (Siebold & Zucc.) Kitag.	Dongdanggui or ribendanggui	Roots	Treatment of menoxenia and anemia	Hemogenic, analgesic, and sedative activities	Ferulic acid, ligustilide, and angelicide	[23]
5	*Angelica amurensis* Schischk.	Heishuidanggui or chaoxiandanggui	Roots	\	\	α-pinene, limonene, and sabinene	[1,24]
6	*Angelica anomala* Avé-Lall.	Xiayedanggui or yixingdanggui	Roots	Dispelling wind, eliminating dampness, and relieving pain	Antioxidant, anti-inflammatory, and antitumor	Isoimperatorin, umbelliferone, and adenosine	[16,25,26,27]
7	*Angelica apaensis* R. H. Shan & C. C. Yuan	Faluohai or abadanggui	Roots	Relieving pain, relieving cough and asthma	Bacteriostat, anti-inflammatory	Oxypeucedanin, isoimperatorin, and oxypeucedanin hydrate	[19,28]
8	** *Angelica biserrata* (R. H. Shan & C. C. Yuan) C. C. Yuan & R. H. Shan	Duhuo or maodanggui	Roots	Dispelling wind, eliminating dampness, and relieving pain	Antitumor, anti-inflammatory, and antioxidant	Coumarins osthole, columbianadin, and volatile oils	[29]
9	*Angelica cartilaginomarginata var. Foliosa* C. C. Yuan & R. H. Shan	Shangaoben	Roots	\	\	\	[17]
10	** *Angelica dahurica* (Fisch. Ex Hoffm.) Benth. & Hook. F. Ex Franch. & Sav.	Baizhi	Roots	Treatment of acne, erythema, and headache	Anti-inflammatory, anti-mutagenic, and antitumor	Scopoletin and psoralen	[18,30,31,32,33]
11	** *Angelica dahurica* cv. Hangbaizhi	Hangbaizhi	Roots	Treatment of headache, toothache, abscess, and furunculosis	Estrogenic, cytotoxic, and anti-inflammatory	Isoimperatorin, imperatorin, and phellopterin	[18,34,35]
12	*Angelica dahurica var. Formosana* (H. Boissieu) Yen	Taiwanduhuo	Roots	\	Anti-staphylococca	Falcarindiol	[33,34]
13	** *Angelica decursiva* (Miq.) Franch. & Sav.	Zihuaqianhu	Roots	A remedy for thick phlegm, asthma, and upper respiratory tract infections	Antioxidant and anti-inflammatory potential	Decursin, decursidin, and nodakenetin	[36]
14	*Angelica gigas* Nakai	Chaoxiandanggui	Roots	Treatment of dysmenorrhea, amenorrhea, and menopause	Anti-platelet effects	Decursin and decursinol angelate	[37,38]
15	*Angelica laxifoliata* Diels	Shuyedanggui	Roots	Dispelling wind and relieving pain	Treatment of wind-damp pain, lumbus, and knees	Angelicin, β-sitosterol, and laxifolin	[16,26,39]
16	*Angelica megaphylla* Diels	Dayedanggui	Roots	Used as *Angelica sinensis*	Used as *A. Sinensis*	Ferulic acid, ligustilide, and angelol	[40,41]
17	*Angelica morii* Hayata	Fushen	Roots and leaves	Treatment of spleen and stomach, cold cough, and toothache	Used for diarrhea caused by deficiency of spleen and for cough caused by weakness and chill	Imperatorin, isoimperatorin, and phellopterin	[42,43,44]
18	*Angelica nitida* H. Wolff	Qinghaidanggui	Roots	Nourishing the blood, regulating menstrual disorder, and relieving pain	\	Isoimperatorin, imperatorin, and cnidilin	[45]
19	*Angelica polymorpha* Maxim.	Guaiqin or shanqincai	Roots	Dispelling wind and relieving pain	Treatment of stomachache	Coumarins, sesquiterpenoids, and alkaloids	[19,46,47]
20	** *Angelica sinensis* (Oliv.) Diels	Danggui	Roots	Nourishing the blood, regulating menstrual disorder, and relieving pain	Cardio-cerebrovascular, anti-inflammatory, and antioxidant	Ferulic acid, alkylphthalides, and polysaccharides	[18,48,49]
21	*Angelica sinensis* var. Wilsonii	Emeidanggui	Roots	Used as *Angelica sinensis*, relieving pain	Used as *Angelica sinensis*	Isoimperatorin, coumarin, and oxypeucedanin	[50]
22	*Angelica sylvestris* L.	Lindanggui	Roots	Relieves rheumatism and sweating, provides detoxification	\	Cnidilide, sedanenolide, and ligustilide	[19]
23	*Angelica tsinlingensis* K. T. Fu	Qinlingdanggui	Roots	\	\	\	[1]
24	*Angelica valida* Diels	Wuduhuo or yandanggui	Roots	\	\	\	[1]
25	*Anthriscus nemorosa* (M. Bieb.) Spreng.	Ciguoeshen	Roots, whole plant, and leaves	Used as *Peucedanum praeruptorum*	Used as *Peucedanum praeruptorum*	\	[51]
26	*Anthriscus sylvestris* (L.) Hoffm.	Eshen	Roots and leaves	Invigorating spleen, replenishing qi, and expelling phlegm	Antitumor, antioxidation, and antisenity	Phenylpropanoids, flavonoids, and steroidal	[19,52]
27	*Apium graveolens* L.	Hanqin	Whole plant, roots, and rhizome	Dispelling wind, eliminating dampness, and detoxification	Hypertension, hyperlipidemia, and dysuria	Organic acids, apigenin, and volatile oils	[19,53,54]
28	*Archangelica brevicaulisf*	Duanjinggudanggui	Roots	Used as *Angelica biserrata*	Used as *Angelica biserrata*	Osthol, imperatorin, and archangelicin	[16,55]
29	*Bupleurum angustissimum* (Franch.) Kitag.	Xiayechaihu	Roots	\	\	Saikosaponins (a, c, and d), β -terpinene, and β -thujene	[56]
30	*Bupleurum aureum* Fisch.	Jinhuangchaihu	Roots	\	\	Saikosaponins (a, c, and d)	[1,57]
31	*Bupleurum bicaule* Helm	Zhuiyechaihu	Roots	Used as *Bupleurum scorzonerifolium*	Used as *Bupleurum scorzonerifolium*	Saikosaponin d, prosaikogenin G, and prosaikogenin F	[16,58,59]
32	*Bupleurum candollei* Wall. Ex DC.	Chuandianchaihu	Whole plant	Diminishing inflammation, detoxification, dispelling wind, and relieving convulsion	\	Saikosaponin and flavonoids	[16,56]
33	*Bupleurum chaishoui* R. H. Shan & M. L. Sheh	Chaishou	Roots and rhizome	Used as *Bupleurum chinense*	Used as *Bupleurum chinense*	Saikosaponins (a, c, and d)	[60]
34	** *Bupleurum chinense* DC.	Beichaihu	Roots	Treatment of chronic hepatitis, kidney syndrome, and inflammatory diseases	Anti-allergen, analgesic, and anti-inflammation	Saikosaponins (a, c, and d)	[18,61,62]
35	*Bupleurum chinense* DC. F. Octoradiatum (Bunge) Shan et Sheh	Baihuashanchaihu	Roots	Used as *Bupleurum chinense*	Anti-allergen, analgesic, and anti-inflammation	Saikosaponins (a, c, and d)	[63,64]
36	*Bupleurum chinense* DC. F. Vanheurckii (Muell.-Arg.) Shan et Y. Li	Yantaichaihu	Roots	Used as *Bupleurum chinense*	Anti-allergen, analgesic, and anti-inflammation	Saikosaponins (a, c, and d)	[63,64]
37	*Bupleurum commelynoideum var. Flaviflorum* R. H. Shan & Yin Li	Huanghuayazhichaihu	Roots, rhizome, and whole plant	Antipyretic-analgesic effect, choleretic, and hepatoprotection	Treating or relieving inflammatory bowel disease	Saikosaponins (a, c, and d), β-pinene, and perillen	[65,66]
38	*Bupleurum densiflorum* Rupr.	Mihuachaihu	Roots	\	\	\	[63]
39	*Bupleurum dielsianum* H. Wolff	Taibaichaihu	Roots	\	\	\	[63]
40	*Bupleurum euphorbioides* Nakai	Dabaochaihu	Roots	\	\	Saikosaponins, perillen, and undecanal	[56]
41	*Bupleurum exaltatum M.* Bieb.	Xinjiangchaihu	Roots	\	\	\	[64]
42	*Bupleurum falcatum* L.	Sandaochaihu	Roots	\	Treatment of colds and upper respiratory tract infections	Saikosaponins (a, c, and d)	[64,67,68]
43	*Bupleurum gansuense* S. L. Pan et Hsu	Gansuchaihu	Roots	\	\	\	[56]
44	*Bupleurum hamiltonii* N. P. Balakr.	Xiaochaihu	Roots or whole plant	Antipyretic-analgesic effect, treatment of chill and fever alternation	Treatment of stomach pain, dysuria, and cough	Kaerophyllin, isokaerophyllin, and ethyl caffeic acid	[69]
45	*Bupleurum hamiltonii var. Hamiltonii*/*Bupleurum tenue*	Xiaochaihu	Roots or whole plant	Used as *Bupleurum hamiltonii* N. P. Balakr.	Used as *Bupleurum hamiltonii* N. P. Balakr.	\	[70]
46	*Bupleurum hamiltonii var. Humile* (Franch.) R. H. Shan & M. L. Sheh	Aixiaochaihu	Roots	\	\	\	[64]
47	*Bupleurum huizei* S. L. Pan sp. Nov.	Huizechaihu	Roots	\	\	\	[64]
48	*Bupleurum kaoi* T. S. Liu, C. Y. Chao & T. I. Chuang	Taiwanchaihu or gaoshichaihu	Roots	\	Treatment of influenza and fever	Saikosaponin a, c	[64]
49	*Bupleurum komarovianum* Lincz.	Changbaichaihu	Roots	Used as *Bupleurum chinense*	Used as *Bupleurum chinense*	Saikosaponins (a, c, and d) and volatile oils (1-caprylene, limonene, and thymol)	[71,72]
50	*Bupleurum krylovianum Schischk. Ex* Krylov	Aertaichaihu	Roots	\	\	Saikosaponins (a, c, and d)	[56,57]
51	*Bupleurum kunmingense* Yin Li & S. L. Pan	Jiuyechaihu	Roots	\	Immunomodulatory	Saikosaponins (a, c, and d), cyclohexanone, and 2- methyldodecane	[56]
52	*Bupleurum longicaule var. Amplexicaule* C. Y. Wu	Baojingchaihu	Roots	\	\	Saikosaponins (a, c, and d)	[64]
53	*Bupleurum longicaule var. Franchetii* H. Boissieu	Kongxinchaihu	Roots or whole plant	\	\	Saikosaponins (a, c, and d), cyclohexanone, and myrcene	[56]
54	*Bupleurum longicaule var. Giraldii* H. Wolff	Qinlingchaihu	Roots	\	\	Saikosaponins (a, c, and d), narcissin, and rutin	[56]
55	*Bupleurum longiradiatum* Turcz.	Dayechaihu	Roots	Treatment of gout and inflammatory illness	Anti-inflammatory and/or antimicrobial	Thymol, butylidene phthalide, and 5-indolol	[73]
56	*Bupleurum luxiense* Yin Li & S. L. Pan	Luxichaihu	Roots	\	\	Saikosaponins (a, c, and d), n-heptaldehyde, and octanal	[56]
57	*Bupleurum malconense* R. H. Shan & Yin Li	Maweichaihu	Whole plant	Hepatoprotection and antipyretic effect	Acute toxicity	Saikosaponins (a, c, and d), rutin, and quercetin	[74,75,76]
58	*Bupleurum marginatum var. Marginatum*	Zichaihu or zhuyefangfeng	Whole plant	Hepatoprotection and antipyretic effect	Anti-allergen, analgesic, and anti-inflammatory	Saikosaponins (a, c, and d), rutin, and quercetin	[74,75,77]
59	*Bupleurum marginatum var. Stenophyllum* (H. Wolff) R. H. Shan & Yin Li	Zhaizhuyechaihu	Whole plant	\	\	Saikosaponins (a, c, and d), chikusaikoside I, II, and 2- methylcyclopentanone	[56]
60	*Bupleurum marginatum* Wall. Ex DC.	Zhuyechaihu	Whole plant and roots	Hepatoprotection and antipyretic effect	Anti-allergen, analgesic, and anti-inflammatory	Saikosaponins (a, c, and d), rutin, and quercetin	[74,75,77]
61	*Bupleurum microcephalum* Diels	Maweichaihu	Whole plant and roots	Hepatoprotection and antipyretic effect	Anti-allergen, analgesic, and anti-inflammatory	Saikosaponins (a, c, and d), rutin, and quercetin	[74,75]
62	*Bupleurum petiolulatum var. tenerum* R. H. Shan & Yin Li	Xijingyoubingchaihu	Whole plant	Antipyretic-analgesic effect	Anti-inflammatory	\	[63,78]
63	*Bupleurum polyclonum* Yin Li & S. L. Pan	Duozhichaihu	Roots	\	Anticancer	Saikosaponins (a, c, and d), 4′-O-saikosaponin-a, and fenchane	[56]
64	*Bupleurum rockii* H. Wolff	Lijiangchaihu	Roots	\	\	Saikosaponins (a, c, and d), thymol, and β-guaiene	[56]
65	*Bupleurum scorzonerifolium* f. Longiradiatum	Changsanhongchaihu	Roots	Used as *Bupleurum chinense*	Used as *Bupleurum chinense*	\	[19]
66	*Bupleurum scorzonerifolium* f. Pauciflorum	Shaohuahongchaihu	Roots	Used as *Bupleurum chinense*	Used as *Bupleurum chinense*	\	[19]
67	*** Bupleurum scorzonerifolium* Willd.	Hongchaihu or zhuyechaihu	Roots	Antipyresis, relief of liver issues and menstrual disorders	Used as *Bupleurum chinense*	Rutin, quercetin, and kaempferol	[18,19]
68	*Bupleurum sibiricum var. Jeholense* (Nakai) Y. C. Chu ex R. H. Shan & Yin Li	Wulingchaihu	Roots	\	\	\	[1]
69	*Bupleurum sibiricum* Vest	Xinganchaihu	Roots	Used as *Bupleurum chinense*	Used as *Bupleurum chinense*	Saikosaponin a, rutin, and quercetin	[16,79,80]
70	*Bupleurum sichuanense* S. L. Pan et Hsu.	Sichuanchaihu	Roots	\	\	Saikosaponins (a, c, and d)	[56]
71	*Bupleurum smithii* H. Wolff	Heichaihu	Roots	Antipyretic-analgesic effect	Anti-inflammatory, immunomodulatory, and anti-hepatic injury	Saponins, volatile oils, and lignans	[81]
72	*Bupleurum smithii var. Parvifolium* R. H. Shan & Yin Li	Xiaoyeheichaihu	Roots	Relief of liver issues and activation of yang energy	Anti-inflammatory, immunomodulatory, and antitumor	Falcarinol, saponins, and flavonoids	[82]
73	*Bupleurum thianschanicum* Freyn	Tianshanchaihu	Roots	\	\	Saikosaponins (a, c, and d)	[57]
74	*Bupleurum triradiatum* Adams ex Hoffm.	Sanfuchaihu	Roots	\	\	\	[1]
75	*Bupleurum wenchuanense* R. H. Shan & Yin Li	Wenchuanchaihu	Roots	Used as *Bupleurum*	Used as *Bupleurum*	Quercetin-3-*O*-α-*L*-rhamnoside, quercetin, and rutin	[16,75]
76	*Bupleurum yinchowense* R. H. Shan & Yin Li	Yinzhouchaihu or hongchaihu	Roots	Antipyresis, relief of liver issues, and activation of yang energy	Used as *Bupleurum*	Saikosaponins (a, c, and d)	[16,65,83,84]
77	*Carum buriaticum* Turcz.	Tiangelvzi	Roots and fruits	\	\	\	[5]
78	*Carum carvi* L.	Zanghuixiang	Roots, fruits, and leaves	Dispelling wind, eliminating dampness, invigorating the stomach, and treating heart disease	Anti-bacterial, antioxidant, and antitumor	Carvone, limonene, and dihydrocarvone	[19,85,86]
79	* *Centella asiatica* (L.) Urb.	Jixuecao	Whole plant	Clearing heat, promoting diuresis, and treating toxicity	Anti-bacterial, anti-depressive, and neuroprotective	Asiaticoside, madecassoside, and elemene	[18,87]
80	** *Changium smyrnioides* H. Wolff	Mingdangshen	Roots	Strengthening with tonics, moistening lungs, clearing phlegm, and calming the liver	Immunomodulatory, relieving fatigue, and enhancing adaptability	Cetylic acid, succinic acid, and imperatorin	[18,88]
81	*Chuanminshen violaceum* M. L. Sheh & R. H. Shan	Chuanmingshen	Roots	Moistening the lungs, clearning phlegm, harmonizing the stomach, and stimulating liquids	Antioxidant, enhancing immunity, and antimutation	Polysaccharides, coumarins, and flavonoids	[89,90,91]
82	*Cicuta virosa* L.	Duqin	Roots and rhizome	Expelling phlegm and detoxification	Treatment of osteomyelitis, gout, and rheumatism	P-cymene, cicutoxine, and L-limonene	[17,92]
83	* *Cnidium monnieri* (L.) Spreng.	Shechuang	Fruits	Dispelling wind, relieving convulsion, treating impotence	Antibacterial, antiviral, and antimutagenesis	Osthole, limonene, and cnidimoside A	[18,93]
84	*Cnidium officinale*	Dongchuanxiong	Roots	Used as *Cnidium monnieri*	Used as *Cnidium monnieri*	\	[1]
85	*Conioselinum acuminatum* (Franch.) Lavrova	Shuigaoben	Roots	\	\	Sabinene, α-pinene, and aromadendrene	[11]
86	*Conioselinum anthriscoides* ‘*Fuxiong*’	Fuxiong	Roots	\	\	β-bergamotene	[11]
87	*Conioselinum tenuisectum* (H. Boissieu) Pimenov & Kljuykov	Xiliegaoben	Roots	\	\	\	[94]
88	*Conioselinum vaginatum* (Spreng.) Thell.	Xinjianggaoben or qiaoshanxiong	Roots	Dispelling wind, eliminating dampness, and relieving pain	Treatment of common cold due to wind-cold and gastro spasm	Diligustilide, daucosterol, and palmitic acid	[19,95]
89	*Conium maculatum* L.	Dushen	Whole plant	Relieving pain and relieving muscular spasm	Treatment of cancer	Coniine, N-methyl-coniine, conhydrine 2-(1-hydroxypropyl)-piperidine	[16,96,97]
90	*Coriandrum sativum* L.	Husui	Whole plant, fruits, and stems	Invigorating the stomach and promoting eruption	Antibacterial, antifungal, and antioxidant	Petroselinic acid, linoleic acid, and oleic acid	[19,98]
91	*Cryptotaenia japonica* Hassk.	Sanyeqin	Whole plant	Treatment of weakness, urinary closure, and swelling	Antioxidant, protection of liver, and anticancer	Friedelin, stigmasterol, and apigenin	[19,99,100]
92	*Cuminum cyminum* L.	Ziranqin	Fruits	Treatment of indigestion and stomach/abdominal pain	Antibacterial, antioxidant, and radical-scavenging properties	α-pinene, 1,8-cineole, and linalool	[19,101]
93	*Cyclorhiza peucedanifolia* (Franch.) Constance	Nanzhuyehuangenqin	Fruits	Enriching the blood, activating the blood, and regulating menstrual disorder	\	\	[102]
94	*Daucus carota* L.	Carrot	Fruits	Treatment of ascariasis, enterobiasis, and tapeworm disease	Insecticidal, anti-bacterial, and anticancer	α-pinene, isophorone oxide, and and quercetrin	[18,103]
95	*Daucus carota* *var. Carota*	Wild carrot	Fruits	Treatment of ascariasis, enterobiasis, and tapeworm disease	Insecticidal, anti-bacterial, and anticancer	α-pinene, β-bisabolene, and luteolin	[18,103]
96	*Daucus carota var. Sativus* Hoffm.	Wild carrot	Roots and basal leaves	Strengthening spleen, treatment of dyspepsia and chronic dysentery	Enhancing immunity, anticancer and anti-aging	Carotene, (1R)-α-pinene, and β-carotene	[19,104]
97	*Eriocycla albescens* (Franch.) H. Wolff	Dianqianghuo	Roots	\	\	\	[1]
98	*Eryngium foetidum* L.	Ciqin	Whole plant	Diuresis, treatment of dropsy and snakebite	Bacteriostat, diminishing of inflammation, and promotion of detumescence	Lanolin alcohol, carotene, and *n*-nonyl aldehyde	[19,105]
99	*Ferula bungeana* Kitag.	Yingawei	Whole plant and seeds	Heat clearing and detoxifying, relieving pain and expelling phlegm, and arresting coughing	Treatment of cold, bronchopneumonia, and pulmonary tuberculosis	Anisole, *d*-fenchone, and limonen	[19,106]
100	*Ferula caspica* M. Bieb.	Lihaiawei	Roots and resin	Eliminating stagnated food, relieving dyspepsia, insecticide	Treatment of toxicity	Umbelliprenin, farnesyl alcohol, and umbelliferone	[107]
101	*Ferula conocaula* Korovin	Yuanzhuijingawei	Resin, roots, and rhizome	Eliminating stagnated food, insecticide, treatment of abdominal mass	Anticancer and treatment of influenza	Umbelliprenin, fezelol, and feterin	[107]
102	*Ferula feruloides* (Steud.) Korovin	Xiangawei	Roots and resin	Treatment of chilliness, and heart and abdominal pain	Insecticidal, bacteriostat, and antitumor	α-pinene, farnesene and toluene	[108,109]
103	** *Ferula fukanensis* K. M. Shen	Fukangawei	Resin	Eliminating stagnated food, relieving dyspepsia, insecticide	Treatment of stomach disease, rheumatism, and joint pain	Ferulic acid, guaiol, and ethyl-p-hydroxybenzoate	[18,19,110,111,112]
104	*Ferula jaeschkeana* Vatke	Zhongyaawei	Resin of overground part	Eliminating stagnated food, insecticide, treatment of tumors, wounds, and peptic ulcers	Antifertility	Jaeschkeanadiol, α-pinene, and β-pinene	[107]
105	*Ferula krylovii* Korovin	Tuoliawei	Resin	Eliminating stagnated food, insecticide	\	Fekrynol, ferukrin and fekrynol acetate	[107]
106	*Ferula lehmannii* Boiss.	Daguoawei	Resin	Detoxification, deodorization, and insecticide	Treatment of gastropathy, rheumatism, and arthralgia	Lehmannolone, sinkianone, and lehmannolone A	[16,113]
107	*Ferula moschata* (Reinsch) Koso-Pol.	Shexiangawei	Roots	Sedative, treatment of spasmolysis and hysteria	Suppresses the replication of human immunodeficiency virus in H9 lymphocytes and suppresses the production of cytokine	Fezelol, fesumtuorin A, and fesumtuorin B	[107]
108	*Ferula olivacea* (Diels) H. Wolff ex Hand.-Mazz.	Lanlvawei	Resin	Wind-heat dispersing, expelling phlegm, and arresting cough	\	\	[16]
109	** *Ferula sinkiangensis* K. M. Shen	Xinjiangawei	Resin	Eliminating stagnated food, detoxification, insecticide	Antioxidant, antitumor, and antiviral	Ferulic acid, fekrynol, and lehmannolone	[16,18,114,115]
110	*Ferula songarica* Pall. Ex Schult.	Zhungaeawei	Resin and whole plant	Eliminating stagnated food, insecticide	\	2,4-dihydroxylacetophenone, 3,3′, 4,4′-biphenyltetracarboxylic acid, and Δ^3^-carene	[116]
111	*Ferula teterrima* Kar. & Kir.	Chouawei	Resin	Eliminating stagnated food, insecticide	Treatment of malaria and dysentery	Feterin, badrakemin, and badrakemin acetate	[116]
112	* *Foeniculum vulgare* Mill.	Xiaohuixiang	Fruits, roots, stems, leaves, and whole plant	Dispelling wind, relieving pain, and harmonizing the stomach	Bacteriostat, anti-inflammatory, and insecticide	Trans-anethole, estragole, and anisaldehyde	[18,19,117,118]
113	** *Glehnia littoralis* F. Schmidt ex Miq.	Beishashen	Roots	Heat clearing and detoxifying, diminishing inflammation, expelling phlegm, and arresting cough	Anti-inflammatory, bacteriostat, and antitumor	Phenyllactic acid, catechol, and quercetin	[18,119]
114	*Hansenia oviformis* (R. H. Shan) Pimenov & Kljuykov	Luanyeqianghuo	Rhizome, roots, and leaves	Treatment of rheumatic arthralgia, cold due to wind-cold, and headache	\	\	[16,102]
115	*Heracleum barmanicum* Kurz	Yinduduhuo	Roots	Treatment of cold abdominalgia	\	\	[16]
116	*Heracleum candicans* Wall. Ex DC.	Baiyunhua	Roots	Dispelling wind, eliminating dampness, and relieving pain	Treatment of cold headache	Bergapten, heraclenin, and imperatorin	[19,120]
117	*Heracleum dissectifolium* K. T. Fu	Duolieduhuo	Roots	Dispelling wind, eliminating dampness, and relieving pain	\	\	[16]
118	*Heracleum fargesii* H. Boissieu	Chengkouduhuo	Roots	\	\	\	[17]
119	*Heracleum franchetii* M. Hiroe	Jianyeduhuo	Roots and rhizome	\	\	\	[121,122]
120	*Heracleum hemsleyanum*	Niuweiduhuo	Roots and rhizome	Dispelling wind, eliminating dampness, and relieving pain	Antioxidant, anti-inflammatory, and antitumor	β-pinene, α-pinene, and (1S)-6,6-dimethyl-2-methylene-bicyclo [3.1.1] heptane	[26,27,123,124]
121	*Heracleum hemsleyanum* Diels	Beiduhuo or dahuo	Roots and rhizome	Dispelling wind, eliminating dampness, and relieving pain	Antioxidant, anti-inflammatory, and antitumor	Osthole, columbianadin, and columbianetin	[26,27]
122	*Heracleum henryi* H. Wolff	Nanguaqi	Roots	Clearing and activating the channels and collaterals, relieving pain and scattered stasis	\	Turgeniifolin B, turgeniifolin C, and bergapten	[125]
123	*Heracleum millefolium var. Millefolium*	Qianyeduhuo or zangdanggui	Roots and rhizome	Detumescence, treating masses, and treating leprosy	\	\	[102,121,122]
124	*Heracleum moellendorffii* Hance	Duanmaoduhuo	Roots and rhizome	Clearing and activating the channels and collaterals, relieving pain and scattered stasis	Bacteriostat	β-pinene, α-pinene, and pentadecane	[123,125,126,127]
125	*Heracleum oreocharis* H. Wolff	Shandiduhuo	Roots	\	\	\	[122]
126	*Heracleum rapula* Franch.	Baiyunhua	Roots	Clearing and activating the channels and collaterals, relieving pain and scattered stasis	Bacteriostat, treatment of asthma, and chronic bronchitis	Ostholce, marmesin, and imperatorin	[19,125,128]
127	*Heracleum scabridum* Franch.	Dianbaizhi or caoduhuo	Roots, rhizome, and fruits	Treatment of common cold due to wind-cold, headache, cough, and asthma	\	Heraclenol, oxypeucedanin-hydrate, and byakangelicin	[129,130,131]
128	*Heracleum souliei* H. Boissieu	Xiaoduhuo	Roots	\	\	Bergapten	[120,122]
129	*Heracleum stenopterum* Diels	Xiachiduhuo	Roots	Treatment of cold and rheumatism	\	Bergapten, isopimpinellin, and sphondin	[16,132]
130	*Heracleum tiliifolium* H. Wolff	Duanyeduhuo	Roots	Dispelling wind, eliminating dampness, and relieving pain	\	\	[16]
131	*Heracleum vicinum* H. Boissieu	Pingjieduhuo	Roots	Used as *Notopterygium incisum*	\	\	[121,122]
132	*Heracleum wenchuanense* F. T. Pu & X. J. He	Wenchuanduhuo	Roots	\	\	\	[122]
133	*Heracleum wolongense* F. T. Pu & X. J. He	Wolongduhuo	Roots	\	\	\	[1,122]
134	*Heracleum yungningense* Hand.-Mazz.	Niuweiduhuo	Roots and rhizome	Treatment of waist and knee pain, limb spasm, and leucoderma	\	Pimpinellin, angelicin, and isobergapten	[26,133]
135	*Hydrocotyle himalaica* P. K. Mukh.	Binghuatianhusui	Whole plant	Heat clearing, detoxifying, and eliminating dampness	\	Asiaticoside, madecassoside, and quercetin	[134,135]
136	*Hydrocotyle hookeri subsp. Chinensis* (Dunn ex R. H. Shan & S. L. Liou) M. F. Watson & M. L. Sheh	Tongqiancao	Whole plant	Relieving pain, diuresis, and removing dampness	Antiviral, antitumor, and anti-bacterial	Flavonoids, triterpenes, and volatile oils	[16,129,135]
137	*Hydrocotyle nepalensis* Hook.	Hongmaticao	Whole plant	Clearing heat and promoting diuresis, dissolving stasis, and hemostasis and detoxification	Antiviral, antitumor, and anti-bacterial	Flavonoids, triterpenes, and volatile oils	[16,135]
138	*Hydrocotyle sibthorpioides* Lam.	Xiaojinqiancao	Whole plant	Heat clearing, diuresis, and detumescence	Anti-ulcer, antilipemic, and antiviral	Quercetin, isorhamnetin, and asiaticoside	[135,136]
139	*Hydrocotyle sibthorpioides var. batrachium* (Hance) Hand.-Mazz. Ex R. H. Shan	Tianhusui or potongqian	Whole plant	Heat clearing and detoxifying, eliminating dampness, and diuresis	Anti-ulcer, spasmolysis, and anti-inflammatory	Benzene propane nitrile, phytol, and caryophyllene oxide	[16,137,138]
140	*Hydrocotyle wilfordii* Maxim.	Yutengcao	Whole plant	As *Hydrocotyle nepalensis* Hook.	As *Hydrocotyle nepalensis* Hook.	Asiaticoside, madecassoside, and quercetin	[134,135]
141	*Hymenidium chloroleucum* (Diels) Pimenov & Kljuykov	Xizangdiangaoben	Roots or whole plant	Regulating flow of qi, invigorating the stomach, and activating blood	Anti-inflammatory, analgesia, and nutritious function	Nobiletin, falcarindiol, and isoliquiritingenin	[19,139,140]
142	*Hymenidium davidii* (Franch.) Pimenov & Kljuykov	Songpanlengziqin	Roots	\	\	\	[141]
143	*Hymenidium delavayi* (Franch.) Pimenov & Kljuykov	Lijianggaoben	Roots	\	\	\	[1,6]
144	*Hymenidium lindleyanum* (Klotzsch) Pimenov & Kljuykov	Tianshanlengziqin	Roots	Treatment of hypertension, coronary heart disease, and altitude stress	\	Bergapten, isoimperatorin, and oxypeucedanin	[142]
145	*Kitagawia formosana* (Hayata) Pimenov	Taiwanqianhu	Roots	\	\	\	[1]
146	*Kitagawia macilenta* (Franch.) Pimenov	Xilieqianhu	Roots	Expelling phlegm	\	\	[143]
147	*Kitagawia terebinthacea* (Fisch. Ex Trevir.) Pimenov	Shifangfeng	Roots	Clearing heat and dispelling wind, calming the adverse-rising energy, and expelling phlegm	Treatment of cold and cough, bronchitis, and cough during pregnancy	Isoepoxybuterixin	[19]
148	*Levisticum officinale* W. D. J. Koch	Oudanggui	Roots	Diuresis, invigorating the stomach, and expelling phlegm	Inhibition of rhythmic uterine contractions, Scavenging oxygen free radicals, and anti-lipid peroxidation	Ligustilide, α-phellandrene, and β-phellandrene	[19,144]
149	*Libanotis buchtormensis* (Fisch.) DC.	Yanfeng	Roots	Treating wind chill, dispelling wind dampness, and relieving pain	Bacteriostat, treatment of common cold due to wind-cold, generalized pain, and cough	Falcarinone, isoimperatorin, and xanthotoxin	[19,145]
150	*Libanotis iliensis* (Lipsky) Korovin	Xiyefangfeng	Roots	Expel wind-cold pathogens, thermolysis, and relieving pain	Treatment of common cold due to wind-cold and rheumatic arthritis	Archangelin and iliensin	[19]
151	*Libanotis lancifolia* K. T. Fu	Yanfeng	Roots	Divergent wind chill, dispelling wind-damp, and relieving pain	Bacteriostat, treatment of common cold due to wind-cold, generalized pain, and cough	Falcarinone, isoimperatorin, and xanthotoxin	[19,145]
152	*Libanotis laticalycina* R. H. Shan & M. L. Sheh	Shuifangfeng	Roots	Dispelling wind, antispasmodic, and relieving pain	Analgesic, sedative, and anti-inflammatory	Octanal, hexanal, and 2-pentylfuran	[16,146,147]
153	*Libanotis seseloides* (Fisch. & C. A. Mey. Ex Turcz.) Turcz.	Xiangqin	Roots	Eliminating dampness, activating spleen, and promote blood circulation	Treatment of obstruction, dysentery, and sores	Edultin	[19]
154	*Libanotis sibirica* (L.) C. A. Mey.	Beixiangqin	Roots	\	\	\	[1]
155	*Libanotis spodotrichoma* K. T. Fu	Changchongqi	Roots	Treating wind chill, dispelling wind dampness, and relieving pain	Bacteriostat, treatment of common cold due to wind-cold, generalized pain, and cough	Falcarinone, isoimperatorin, and xanthotoxin	[19,145]
156	*Ligusticopsis brachyloba* (Franch.) Leute	Maoqianhu	Roots	Sudation, relieving pain, and dispelling wind	Treatment of headache, dizziness, arthralgia, and tetanus	α-pinene, β-pinene, and sabinene	[148,149,150]
157	*Ligusticopsis daucoides* (Franch.) Lavrova & Kljuykov	Yubaogaoben	Roots	\	\	\	[1,94]
158	*Ligusticopsis likiangensis* (H. Wolff) Lavrova & Kljuykov	Meimaigaoben	Roots	\	\	\	[1,94]
159	** *Ligusticum chuanxiong* Hort.	Chuanxiong	Roots, rhizome, stems, and leaves	Activating blood, relieving pain, and dispelling wind	Anti-inflammatory, antioxidant, and antitumor	Abietene, tetramethylpyrazine, and glucose	[18,19,151]
160	** *Ligusticum jeholense* Nakai et Kitag.	Liaogaoben	Roots and rhizome	Dispelling wind, dispersing cold, and eliminating dampness	Anti-inflammatory, sedative, and anti-ulcer	Ferulic acid, isoferulic acid, and daucosterol	[18,19,152,153]
161	*Ligusticum pteridophyllum* Franch.	Jueyegaoben	Roots	Dispelling wind, relieving pain, and eliminating dampness	Treatment of cold due to wind-cold and migraine	Asaricin, β-sitosterol, and daucosterol	[26,154]
162	** *Ligusticum sinense* Oliv.	Baogen	Roots, rhizome, and tuber	Expelling wind-cold pathogens, eliminating dampness, and relieving pain	Anti-inflammatory, central inhibitory, and anti-thrombotic effects	3-butylphthalide, opthalonide, and neopthalonide	[18,155]
163	*Ligusticum tenuissimum* (Nakai) Kitagawa	Gaoben	Roots and rhizome	Used as *ligusticum sinense* Oliv. Treatment of wind chill, wind-cold headache, and diarrhea	Analgesia and sedation	Ferulic acid	[19,94,156]
164	*Meeboldia delavayi* (Franch.) W. Gou & X. J. He	Dianqin	Roots	Treatment of cold, fever, and headache	\	\	[16]
165	*Nothosmyrnium japonicum var. Japonicum*	Baibaoqin	Roots	\	Sedation and analgesia	\	[16]
166	*Nothosmyrnium japonicum var. Sutchuensis* H. Boissieu	Chuanbaibaoqin	Roots	\	Sedation and analgesia	\	[16]
167	** *Notopterygium franchetii* H. De Boiss.	Kuanyeqianghuo	Roots and rhizome	Treating wind chill, dispelling wind, and eliminating dampness	Anti-inflammatory, analgesic, and antiviral	Nodakenin, ferulic acid, and bergamot lactone	[18,157,158]
168	** *Notopterygium incisum* Ting ex H. T. Chang	Qianghuo	Roots and rhizome	Treating wind chill, dispelling wind, and eliminating dampness	Anti-inflammatory, analgesic, and antiviral	Nodakenin, notopterol, and isoimperatorin	[18,158]
169	*Oenanthe benghalensis* Benth. & Hook.	Shaohuashuiqin	Roots and whole plant	Used as *Oenanthe javanica* (Blume) DC.	Used as *Oenanthe javanica* (Blume) DC.	\	[17,159]
170	*Oenanthe javanica* (Blume) DC.	Shuiqin	Roots, stems, and whole plant	Heat clearing, detoxification, and removing liver-fire	Enhancing immunity, antiarrhythmic, and hypoglycemic	Phytic acid, γ-terpinene, and caryophyllene	[19,160]
171	*Oenanthe linearis subsp. Rivularis* (Dunn) C. Y. Wu & F. T. Pu	Yeshuiqin	Roots and whole plant	Used as *Oenanthe javanica* (Blume) DC.	Used as *Oenanthe javanica* (Blume) DC.	\	[17]
172	*Osmorhiza aristata var. Laxa* (Royle) Constance & R. H. Shan	Xianggenqin	Roots	Treating wind chill and sudation, and relieving pain	\	\	[16]
173	*Ostericum citriodorum* (Hance) C. C. Yuan & R. H. Shan	Geshanxiang	Roots and whole plant	Activating blood, dissolving stasis, and dispelling wind	Expectorant, anti-inflammatory, and bacteriostat	Isoapiole, panaxynol, and myristicin	[19,161,162,163]
174	*Ostericum grosseserratum* (Maxim.) Kitag.	Dachishanqin	Roots	Activating spleen, dispersing cold, invigorating spleen, and replenishing qi	\	Octanal, β-pinene, and myristic acid	[16,164,165]
175	*Ostericum sieboldii* (Miq.) Nakai	Shanqin	Roots	\	\	\	[166,167,168]
176	*Peucedanum dielsianum* Fedde ex H. Wolff	Chuanfangfeng	Roots and rhizome	Relieving pain, dispelling wind, and eliminating dampness	\	Isoimperatorin, Phellopterin, and 9-octadecenoic acid	[19,169,170]
177	*Peucedanum dissolutum* (Diels) H. Wolff	Yanfeng	Roots	\	\	\	[1]
178	*Peucedanum harry-smithii var. Subglabrum*	Yingqianhu	Roots	Used as *Peucedanum praeruptorum*; alleviating asthma, reducing phlegm, and heat elimination	Treatment of bronchitis, hypertension, and coronary heart disease	Psoralen, bergapten, and xanthotoxin	[171,172,173,174]
179	*Peucedanum japonicum* Thunb.	Binhaiqianhu	Roots	Clearing heat, relieving cough, and diuresis	Antipyresis, analgesia, and anti-inflammatory	Peucedanol, umbelliferone, and β-pinene	[19,175,176]
180	*Peucedanum ledebourielloides* K. T. Fu	Huashanqianhu	Roots	\	\	\	[1,168]
181	*Peucedanum longshengense* R. H. Shan & M. L. Sheh	Nanlingqianhu	Roots	\	\	\	[1]
182	*Peucedanum mashanense* R. H. Shan & M. L. Sheh	Fangfeng	Roots	Expelling phlegm	\	\	[143]
183	*Peucedanum medicum* Dunn	Huazhongqianhu	Roots	Expelling phlegm, alleviating asthma and cough, and arresting convulsion	Anticoagulation, antioxidant, and antibacterial	2-methoxy-4-vinylphenol, *p*-menthan-1-ol, and *cis*-α-bisabolene	[19,177,178]
184	*Peucedanum medicum var. Gracile* Dunn ex R. H. Shan & M. L. Sheh	Yanqianhu	Roots and rhizome	Expelling phlegm, alleviating asthma and cough, and arresting convulsion	Anticoagulation, antioxidant, and antibacterial	Isoimperatorin, phellorerin, and bergapten	[19,177,179]
185	*Peucedanum medicum var. Medicum*	Huazhongqianhu	Roots and rhizome	Expelling phlegm, alleviating asthma and cough, and arresting convulsion	Anticoagulation, antioxidant, and antibacterial	2-methoxy-4-vinylphenol, *p*-menthan-1-ol, and *cis*-α-bisabolene	[19,177,178]
186	** *Peucedanum praeruptorum* Dunn	Qianhu	Roots	Treating gas, clearing heat, and reducing phlegm	Anticoagulation, antioxidant, and anticancer	Praeruptorin A, praeruptorin B, and scopoletin	[18,180]
187	*Peucedanum shanianum* F. L. Chen & Y. F. Deng	Hongqianhu	Roots	Relieving asthma, expelling phlegm, and treating spasmolysis	Anti-inflammatory, antiallergic, and anti-ulcer	Sinodielides A, deltoin, and (+)-pareruptorin A	[181,182,183,184]
188	*Peucedanum turgeniifolium* H. Wolff/*Peucedanum pulchrum*	Yaqianhu	Roots and whole plant	Expelling phlegm, antibechic, and dispersing wind-heat	Smooth muscle spasmolysis	Turgenifolin A, turgenifolin B, and bergapten	[19,184,185]
189	*Peucedanum wawrae* (H. Wolff) S. W. Su ex M. L. Sheh	Taishanqianhu	Roots	Antibechic and expelling phlegm	Analgesia, sedation, and anti-inflammatory	Peucedanocoumarin, d-laserpitin, and bergapten	[16,168,186]
190	*Peucedanum wulongense* R. H. Shan & M. L. Sheh	Wulongqianhu	Roots	\	\	\	[1]
191	*Phlojodicarpus sibiricus* (Steph. Ex Spreng.) Koso-Pol.	Zhangguoqin	Roots	\	\	\	[1]
192	*Physospermopsis alepidioides* (H. Wolff & Hand.-Mazz.) R. H. Shan	Quanyedianxiong	Roots	\	\	\	[1]
193	*Physospermopsis delavayi* (Franch.) H. Wolff	Dianxiong	Roots	\	\	\	[1]
194	*Pimpinella anisum* L.	Huiqin	Fruits	Warming meridian and diuresis	Treatment of paralysis, facial paralysis, and migraine	Anisaldehyde, anisole, and (E)-anethole	[187,188,189,190,191]
195	*Pimpinella candolleana* Wight & Arn.	Xingyefangfeng	Roots or whole plant	Warming spleen and stomach for dispelling cold, relieving pain, and dispelling wind	Relieving muscular spasm, antiviral, and antibacterial	α-zingiberene, pregeijerene, and β-elemene	[19,192,193,194]
196	*Pimpinella coriacea* (Franch.) H. Boissieu	Geyehuiqin	Whole plant	Warming spleen and stomach for dispelling cold, dispelling wind, and eliminating dampness, and activating blood	\	\	[195]
197	*Pimpinella diversifolia* DC.	Yiyehuiqin	Whole plant	Expelling phlegm, activating blood, relieving pain, and removing toxicity for detumescence	Anti-inflammatory, antitumor, and anti-tuberculosis	1H-benzocycloheptene, sesquiphellandrene, and β-chamigrene	[196,197,198]
198	*Pimpinella diversifolia var. Diversifolia*	Yiyehuiqin	Roots or whole plant	Invigorating stomach, dispersing accumulations, and antidiarrheic	Anti-inflammatory, antitumor, and anti-tuberculosis	1H-benzocycloheptene, sesquiphellandrene, and β-chamigrene	[19,196,197,198]
199	*Pimpinella thellungiana* H. Wolff	Yanghongshan	Roots or whole plant	Warming spleen and stomach for dispelling cold, benefiting qi and nourishing blood, and coordinating yin and yang	Hypotensive, hypolipidemic, and modulates and improves cellular immunity	Protocatechuic acid, gallic acid, and neochlorogenic acid	[199,200,201,202,203]
200	*Pleurospermopsis bicolor* (Franch.) Jing Zhou & J. Wei	erselengziqin	Whole plant	Warming spleen and stomach for dispelling cold, benefiting qi and nourishing blood, and coordinating yin and yang	Hypotensive, antilipemic, and modulates and improves cellular immunity, antimicrobial	Chlorogenic acid, isochlorogenic acid A, and apigenin-7-*O*-β-D-glucuronopyranoside	[199,201,202]
201	*Pleurospermum aromaticum* W. W. Sm.	fangxianglengziqin	Whole plant	\	\	\	[1]
202	*Pleurospermum giraldii* Diels	Taibaidiangaoben	Whole plant and seeds	Warming spleen, digesting food, and treating vaginal discharge	Inhibition of smooth muscle contraction and releasing intestinal smooth muscle spasm	Carvone, n-triactanol, and γ-sitosterol	[19,204,205,206]
203	*Pleurospermum rivulorum* (Diels) K. T. Fu & Y. C. Ho	Shetouqianghuo	Roots or whole plant	Tonifying the kidney	\	\	[1,102]
204	*Pternopetalum leptophyllum* (Dunn) Hand.-Mazz.	Baoyenangbanqin	Whole plant	\	\	\	[16]
205	*Pternopetalum vulgare var. Vulgare*	Wupiqing	Roots or whole plant	Treatment of lumbago	\	\	[19]
206	*Sanicula astrantiifolia* H. Wolff ex Kretschmer	Wupifeng or xiaoheiyao	Whole plant	Tonifying the kidney and lung, treating tuberculosis and kidney vacuity lumbar pain	Antioxidant, antibacterial, and bacteriostat	Total flavonoids, rutin, and polysaccharides	[207,208,209]
207	*Sanicula caerulescens* Franch.	Dafeijincao	Whole plant	Dispelling wind, treating phlegm, and promoting blood circulation for regulating menstruation	Expectorant, antibechic, and anti-inflammatory	Angelicin, isoferulaldehyde, and 12-hydroxybakuchiol	[19,210,211]
208	*Sanicula chinensis* Bunge	Shanqincai	Whole plant	Detoxification, hemostasis, and treatment of throat pain	Antiviral	\	[129,212,213,214]
209	*Sanicula elata* Buch.-Ham. Ex D. Don	Sanyeqi	Whole plant	Used as *Sanicula lamelligera*	Antiviral	Oleanane saponins, saponins, and microelement	[212,213,214,215,216,217]
210	*Sanicula lamelligera* Hance	Dafeijincao	Whole plant	Dispelling wind, treating phlegm, and promoting blood circulation for regulating menstruation	Expectorant, antibechic, and anti-inflammatory	Angelicin, isoferulaldehyde, and 12-hydroxybakuchiol	[19,210,211]
211	*Sanicula orthacantha* S. Moore	Heiejiaoban	Roots or whole plant	Heat clearing and detoxifying, treatment of traumatic injury	\	\	[16]
212	*Sanicula orthacantha var. Brevispina* H. Boissieu	Yajiaoqi	Whole plant	Heat clearing and detoxifying, treatment of traumatic injury	\	\	[16]
213	** *Saposhnikovia divaricata* (Turcz.) Schischk.	Fangfeng	Roots	Dispelling wind, removing dampness to relieve pain, and arresting convulsion	Analgesia, sedation, and anti-inflammatory	Prim-*o*-glucosylcimifugin, 5-*O*-methylvisamitol glycoside, and cimifugin	[18,218,219]
214	*Selinum cryptotaenium* H. Boissieu	Linagshechuang	Roots	\	\	\	[1]
215	*Semenovia montana* Kamelin & V. M. Vinogr.	Lieyeduhuo	Roots	\	\	\	[122]
216	*Seseli delavayi* Franch.	Yunfangfeng	Roots	Dispelling wind, removing dampness, and relieving pain	\	\	[19]
217	*Seseli mairei var. Mairei*	Yunfangfeng	Roots and rhizome	Dispelling wind, removing dampness, and relieving pain	Antipyretic, analgesic, and anti-inflammatory	Sphondin, bergapten, and isopimpinellin	[19,220,221,222]
218	*Seseli yunnanense* Franch.	Chuanfangfeng	Roots and rhizome	Dispelling wind, removing dampness, and relieving pain	Antipyretic, analgesic, and anti-inflammatory	Falcarindiol, falcarinol, and glycerol monolinoleate	[19,220,221,223]
219	*Seselopsis tianschanica* Schischk.	Xiguiqin	Roots	Treatment of fall injury, anemia, and other diseases	Treatment of nasopharynx cancer	\	[16]
220	*Sium suave* Walter	Caogaoben	Whole plant	Dispersing cold, relieving headache, and decreasing blood pressure	\	\	[16,224]
221	*Spuriopimpinella arguta* (Diels) X. J. He & Z. X. Wang	Jianchidayeqin	Roots and whole plant	\	\	\	[195]
222	*Tongoloa silaifolia* (H. Boissieu) H. Wolff	Taibaisanqi	Roots	Stopping bleeding, relieving pain, and activating blood	Treatment of traumatic injury, trauma bleeding, and rheumatic pain	Suberosin, crenulatin, and isoimperatorin	[19,225,226]
223	*Tongoloa stewardii* H. Wolff	Gulingdongeqin	Roots	\	\	\	[1]
224	*Torilis japonica* (Houtt.) DC.	Heshi	Fruits and roots	Lumbricide ascaricide and external antiphlogistic agent	\	Essential oil	[19]
225	*Torilis scabra* (Thunb.) DC.	Huananheshi	Fruits or whole plant	Activating blood, insecticide, and antidiarrheal	Bacteriostat	Cyclohexene, 6,6-dimethyl-bicyclo [3.1.1] heptane-2-carboxaldehyde, and endo-borneol	[19,195,227]
226	*Trachyspermum ammi* (L.) Sprague.	Ayuwei	Fruits	Dispersing cold, relieving pain, and treating indigestion	Antibacterial, antimicrobial, and antifungal	thymol, ρ-cymene, and β-pinene	[19,188,228,229,230,231]
227	*Vicatia thibetica* H. Boissieu	Xigui	Roots	Dispelling wind, eliminating dampness, and dispelling cold	Anti-fatigue, antioxidant, and enhancing immunity	Umbelliferone, bergapten, and ferulic acid	[232,233,234]
228	*Visnaga daucoides* Gaertn.	Amiqin	Fruits	Treatment of coronary artery disease, such as coronary thrombosis	Treatment of renal colic, angina pectoris, and urinary calculi	Khellin, visnagin, and khellol glycoside	[16,235]

Note: * means the plant reported in “*Pharmacopoeia of the People’s Republic of China* (2020)”, ** means the plant roots used as medicine reported in “*Pharmacopoeia of the People’s Republic of China* (2020)” [18].

**Table 2 molecules-28-04384-t002:** Quality markers in the 22 AMPs recorded in the “*Pharmacopoeia of the People’s Republic of China”* (2020) [18].

No./No. in Table 1	Plant Species	Quality Markers	Classification	Biosynthetic Pathway
1/8	*Angelica biserrata*	Osthole (1) and columbianadin (2)	Coumarins	Phenylpropanoids
2/10	*Angelica dahurica*	Imperatorin (3) and isoimperatorin (4)	Coumarins	Phenylpropanoids
3/11	*Angelica dahurica* cv. Hangbaizhi	(3) and (4)	Coumarins	Phenylpropanoids
4/13	*Angelica decursiva*	Nodakenin (5)	Coumarins	Phenylpropanoids
5/20	*Angelica sinensis*	Ferulic acid (6) and ligustilide (15)	Propenyl benzenes and phthalides	Phenylpropanoids and phthalides
6/34	*Bupleurum chinense*	Saikosaponin a (11) and saikosaponin d (12)	Triterpenes	Terpenes
7/67	*Bupleurum scorzonerifolium*	(11) and (12)	Triterpenes	Terpenes
8/79	*Changium asiatica*	Asiaticoside (13) and madecassoside (14)	Triterpenes	Terpenes
9/80	*Changium smyrnioides*	–	–	–
10/83	*Changium monnieri*	(1)	Coumarins	Phenylpropanoids
11/94	*Daucus carota*	–	–	–
12/102	*Ferula fukanensis*	–	–	–
13/109	*Ferula sinkiangensis*	–	–	–
14/112	*Foeniculum vulgare*	Trans-anethole (7)	Phenylpropene	Phenylpropanoids
15/113	*Glehnia littoralis*	–	–	–
16/159	*Ligusticum chuanxiong*	(6) and levistilide A (16)	Phenylpropanoids and phthalide	Phenylpropanoids and phthalides
17/160	*Ligusticum jeholense*	(6)	Phenylpropanoids	Phenylpropanoids
18/162	*Ligusticum sinense*	(6)	Phenylpropanoids	Phenylpropanoids
19/167	*Notopterygium franchetii*	(4), (5), and notopterol (8)	Coumarins	Phenylpropanoids
20/168	*Notopterygium incisum*	(4), (5), and (8)	Coumarins	Phenylpropanoids
21/186	*Peucedanum praeruptorum*	Praeruptorin A (9) and praeruptorin B (10)	Coumarins	Phenylpropanoids
22/213	*Saposhnikovia divaricata*	Prim-*O*-glucosylcimifugin (17) and 5-*O*-methylvisammioside (18)	Chromones	Chromones

Note: “–” indicates there are no specific quality markers recorded in the “*Pharmacopoeia of the People’s Republic of China”* (2020) [18].

**Table 3 molecules-28-04384-t003:** Classification of the 38 rhizomatous AMPs affected by BF.

No./No. in Table 1	Plant Species	Classes	References	No./No. in Table 1	Plant Species	Classes	References
1/4	*Angelica acutiloba* (Siebold & Zucc.) Kitag.	(1)	[306]	20/109	* *Ferula sinkiangensis* K. M. Shen	(3)	[19]
2/8	** *Angelica biserrata* (R. H. Shan & C. C. Yuan) C. C. Yuan & R. H. Shan	(1)	[307]	21/111	*Ferula teterrima* Kar. & Kir.	(3)	[19]
3/10	** *Angelica dahurica* (Fisch. ex Hoffm.) Benth. & Hook. f. ex Franch. & Sav.	(1)	[308]	22/113	** *Glehnia littoralis* F. Schmidt ex Miq.	(2)	[309]
4/11	** *Angelica dahurica cv. Hangbaizhi*	(1)	[308]	23/121	*Heracleum hemsleyanum* Diels	(1)	[307]
5/13	** *Angelica decursiva* (Miq.) Franch. & Sav.	(1)	[310]	24/126	*Heracleum rapula* Franch.	(1)	[19]
6/14	*Angelica gigas* Nakai	(2)	[311]	25/148	*Levisticum officinale* W. D. J. Koch	(3)	[19]
7/19	*Angelica polymorpha* Maxim.	(1)	[19]	26/149	*Libanotis buchtormensis* (Fisch.) DC	(3)	[312]
8/20	** *Angelica sinensis* (Oliv.) Diels	(1)	[313]	27/150	*Libanotis iliensis* (Lipsky) Korovin	(1)	[19]
9/26	*Anthriscus sylvestris* (L.) Hoffm.	(3)	[314]	28/151	*Libanotis lancifolia* K. T. Fu	(3)	[19,312]
10/34	** *Bupleurum chinense* DC.	(2)	[315]	29/153	*Libanotis seseloides* (Fisch. & C. A. Mey. ex Turcz.) Turcz.	(1)	[19]
11/67	** *Bupleurum scorzonerifolium* Willd.	(2)	[315]	30/155	*Libanotis spodotrichoma* K. T. Fu	(3)	[19,312]
12/80	** *Changium smyrnioides* H. Wolff	(2)	[316]	31/159	** *Ligusticum chuanxiong* Hort.	(2)	[317]
13/81	*Chuanminshen violaceum* M. L. Sheh & R. H. Shan	(2)	[318]	32/160	** *Ligusticum jeholense* Nakai et Kitag.	(2)	[319]
14/82	*Cicuta virosa* L.	(3)	[19]	33/162	** *Ligusticum sinense* Oliv.	(2)	[319]
15/96	*Daucus carota var. sativus* Hoffm.	(1)	[320]	34/167	** *Notopterygium franchetii* H. de Boiss.	(2)	[321]
16/102	*Ferula feruloides* (Steud.) Korovin	(3)	[19]	35/168	** *Notopterygium incisum* Ting ex H. T. Chang	(2)	[321]
17/103	*Ferula fukanensis* K. M. Shen	(3)	[19]	36/186	** *Peucedanum praeruptorum* Dunn	(1)	[322]
18/106	*Ferula lehmannii* Boiss.	(3)	[19]	37/195	*Pimpinella candolleana* Wight & Arn.	(3)	[19]
19/108	*Ferula olivacea* (Diels) H. Wolff ex Hand.-Mazz.	(3)	[19]	38/213	** *Saposhnikovia divaricata* (Turcz.) Schischk.	(1)	[323,324]

Note: (1) BF significantly affects the yield and quality, and the rhizomes or roots cannot be used for clinical applications; (2) BF differently affects the yield, but the rhizomes or roots can be used as medicine to some extent; and (3) BF has no significant effect on the yield and quality, and their rhizomes or roots are used as medicine. * means the plant reported in “*Pharmacopoeia of the People’s Republic of China* (2020*)*”, ** means the plant roots used as medicine reported in “*Pharmacopoeia of the People’s Republic of China* (2020)” [18].

**Table 4 molecules-28-04384-t004:** Reported approaches for inhibiting BF in 25 AMPs.

Class	No./No. in Table 1	Plant Species	Measure I (Seeding)	Measure II (Cultivation)	Measure III (Abiotic)	Measure IV (Molecular Biology)
(1)	1/4	*Angelica acutiloba* (Siebold & Zucc.) Kitag.	Seedling diameter [339]	Density of planted seedlings [339]	Paclobutrazol concentration [339]	\
(1)	2/8	** *Angelica biserrate* (R. H. Shan & C. C. Yuan) C. C. Yuan & R. H. Shan	Seedling size and root length [307]	\	\	\
(1)	3/10	** *Angelica dahurica* (Fisch. ex Hoffm.) Benth. & Hook. f. ex Franch. & Sav.	Seed quality and seed maturity degree [308,333]	Soil selection to avoid continuous cropping and fertile sticky soil, density of planted seedlings, and seeding time [333,335,340]	Rational application of fertilizer, and appropriate N, P, and K fertilizer [308,333,341]	Seven types of reproductive conversion genes, and constans-like genes [342,343]
(1)	4/11	** *Angelica dahurica* cv. Hangbaizhi	Seed quality and seed maturity degree [308,333]	Soil selection to avoid continuous cropping and fertile sticky soil, density of planted seedlings, and seeding time [333,335,340]	Rational application of fertilizer, and appropriate N, P, and K fertilizer [308,333,341]	Seven types of reproductive conversion genes, and constans-like genes [342,343]
(1)	5/13	** *Angelica decursiva* (Miq.) Franch. & Sav.	\	\	\	\
(1)	6/19	*Angelica polymorpha* Maxim.	\	\	\	\
(1)	7/20	** *Angelica sinensis* (Oliv.) Diels	Seed maturity degree, seeding age, seeding weight, root diameter, and excellent variety [328,330,344,345,346]	Short day, storage temperature, and reasonable planting and cultivation [313,344,347]	Plant growth retardant [348]	Four pathways of genes for regulating early BF [349,350]
(1)	8/96	*Daucus carota var. Sativus* Hoffm.	Endogenous hormone content and different cultivars [351,352]	Temperature, short day, and seeding time [351,353,354,355]	\	Two major genes: *Bol1-1* and *Bol1-2* [356]
(1)	9/121	*Heracleum hemsleyanum* Diels	\	\	\	\
(1)	10/126	*Heracleum rapula* Franch.	\	\	\	\
(1)	11/150	*Libanotis iliensis* (Lipsky) Korovin	\	\	\	\
(1)	12/153	*Libanotis seseloides* (Fisch. & C. A. Mey. ex Turcz.) Turcz.	\	\	\	\
(1)	13/186	** *Peucedanum praeruptorum* Dunn	\	Compact planting and seeding time [357,358]	\	\
(1)	14/213	** *Saposhnikovia divaricata* (Turcz.) Schischk.	\	Density of planted seedlings [338]	\	Differentially expressed genes associated with BF during early flowering, flower bud differentiation, and late flowering [359]
(2)	15/14	*Angelica gigas* Nakai	\	\	\	\
(2)	16/34	** *Bupleurum chinense* DC.	\	Cut the flowers [315]	Temperature [360]	Flowering gene (BcSVP, BcPAF1, BcCO, and BcFT) [361]
(2)	17/67	** *Bupleurum scorzonerifolium* Willd.	\	\	\	\
(2)	18/80	*Changium smyrnioides* H. Wolff	\	Cut the flowers [316]	\	\
(2)	19/81	*Chuanminshen violaceum* M. L. Sheh & R. H. Shan	\	\	\	\
(2)	20/113	** *Glehnia littoralis* F. Schmidt ex Miq.	\	Cut the flowers [309]	\	\
(2)	21/159	** *Ligusticum chuanxiong* Hort.	Asexual reproduction and tissue culture [317,362]	Cut the bolted stem [363]	\	Transcriptome original data by Illumina sequencing technology [364]
(2)	22/160	** *Ligusticum jeholense* Nakai et Kitag.	\	Cut the flower [365,366]	\	\
(2)	23/162	** *Ligusticum sinense* Oliv.	\	Cut the flower [365,366]	\	\
(2)	24/167	** *Notopterygium franchetii* H. de Boiss.	\	Cut the flower [367]	\	\
(2)	25/168	** *Notopterygium incisum* Ting ex H. T. Chang	\	Cut the flower [321]	\	\

Note: ** means the plant roots used as medicine reported in “*Pharmacopoeia of the People’s Republic of China* (2020)” [18].

## Data Availability

Not applicable.

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
