# Peer review of "Apiaceae Medicinal Plants in China: A Review of Traditional Uses, Phytochemistry, Bolting and Flowering (BF), and BF Control Methods"

_molecules, 2023, doi:10.3390/molecules28114384_

Round 1

Reviewer 1 Report

Though it is an interesting review article, which can be considered due to its importance in the field. I have gone through the manuscript. It is well-written, thorough and in-depth. However, there are some places which needs further corrections. 

Firstly, thorough English editing is required. Please revise the manuscript taking help from a colleague who is proficient in English and familiar with the subject matter, who can review your manuscript, or contact a professional editing service to review your manuscript.  Lot of typo errors are there. For example see the line 4 in abstract: ....... improve the yield and quality of Apiaceae medicinal plants (AMPs), The traditional use, phytochemistry.......

"T" should be small in The

Secondly, revise the title. This review is not generally written on Apiaceae family. It is completely focussed on Chinese medicine. Therefore, I suggest authors to revise the title to "Apiaceae medicinal plants in Chinese medicine: A review of traditional uses, phytochemistry, bolting and flowering, and controlling approaches.

In table 1, I suggest authors to add another column next to plant species, and write the local names of all plant species used in Chinese medicine. It will be very easier in identification and for keywords search.

I believe, figure 2 and 3 requires references either in text or in legends.

English language, typos, grammatical errors are lot and significant improvement is required.

Author Response

Though it is an interesting review article, which can be considered due to its importance in the field. I have gone through the manuscript. It is well-written, thorough and in-depth. However, there are some places which need further corrections.

Firstly, thorough English editing is required. Please revise the manuscript taking help from a colleague who is proficient in English and familiar with the subject matter, who can review your manuscript, or contact a professional editing service to review your manuscript. Lot of typo errors are there. For example, see the line 4 in abstract: ....... improve the yield and quality of Apiaceae medicinal plants (AMPs), The traditional use, phytochemistry......."T" should be small in The.

Thanks for your suggestion, the word "The" has been corrected to “the”. (Page 1, line 17) In addition, the English throughout the text has been carefully checked to avoid the errors.

Secondly, revise the title. This review is not generally written on Apiaceae family. It is completely focused on Chinese medicine. Therefore, I suggest authors to revise the title to "Apiaceae medicinal plants in Chinese medicine: A review of traditional uses, phytochemistry, bolting and flowering, and controlling approaches.

Thanks for your suggestion, the title has been revised to “Apiaceae medicinal plants in China: A review of traditional uses, phytochemistry, bolting and flowering, and controlling approaches.” (Page 1, line 2)

In Table 1, I suggest authors to add another column next to plant species, and write the local names of all plant species used in Chinese medicine. It will be very easier in identification and for keywords search.

Thanks for your suggestion, the column of local names in Chinese has been added next to plant species in Table 1. (Page 3, line 90)

I believe, Figure 2 and 3 requires references either in text or in legends.

Thanks for your suggestion, Figure 2 and 3 have cited references in Table 1. (Page 3, line 90)

Reviewer 2 Report

The manuscript summarizes Apiaceae medicinal plants the progress on traditional use, phytochemistry, bolting and flowering, and controlling approaches were summarized.  This reviews will provide references for efficient cultivation and quality improvement of AMP. The author reviewed the traditional efficacy, chemical composition and application of Apiaceae medicinal plants from various aspects. The overall content was very substantial with clear content, clear structure and high completion quality. After careful review, some detailed review comments are listed below

1. This manuscript has a clear hierarchy, but the structure is slightly inadequate.  The authors should be unified about abbreviation.

2. It is recommended that authors use special marks when quoting classic books.

3.The author should briefly describe the pharmacological effects of Apiaceae medicinal plants.

4. The authors are advised to revise the manuscript against English language errors that are present throughout the manuscript.

The authors are advised to revise the manuscript against English language errors that are present throughout the manuscript.

Author Response

The manuscript summarizes Apiaceae medicinal plants the progress on traditional use, phytochemistry, bolting and flowering, and controlling approaches were summarized.  This reviews will provide references for efficient cultivation and quality improvement of AMP. The author reviewed the traditional efficacy, chemical composition and application of Apiaceae medicinal plants from various aspects. The overall content was very substantial with clear content, clear structure and high completion quality. After careful review, some detailed review comments are listed below.

1. This manuscript has a clear hierarchy, but the structure is slightly inadequate. The authors should be unified about abbreviation.

Thanks for your suggestion, all abbreviations have been unified in the manuscripts, for example, bolting and flowering (BF).

2. It is recommended that authors use special marks when quoting classic books.

Thanks for your suggestion, the classic books, like Pharmacopoeia of the People’s Republic of China (2020), have been quoted with “*” in Tables 1 and 3. (Page 3, line 90 and Page 23, line 372) In addition, the classic books have been quoted with “” in the manuscripts. (Page 2, lines 68 to 73; Page 2, lines 77 to 87)

3. The author should briefly describe the pharmacological effects of Apiaceae medicinal plants.

Thanks for your suggestion, the pharmacological effects of Apiaceae medicinal plants have been briefly described. (Page16, lines 161 to 164)

4. The authors are advised to revise the manuscript against English language errors that are present throughout the manuscript.

According to your comments, the English throughout the text has been carefully checked to avoid the errors.

Reviewer 3 Report

The manuscript is well organized and brings inside into the traditional and modern use, secondary metabolites and systematic approach for controlling the bolting and flowering of Apiaceae species in TCM.

Some minor concerns arise:

1.    As the discussed plants are related to the TCM, the last should be embedded into the title

2.    The term secondary (specialized) metabolites should be embedded into the manuscript (kinds of metabolites should be omitted).

3.    Table 1. Rephrase the table title in more explicating manner. Concerning N 35, 36, 65, 66, 69 etc., “same as Bupleurum” is mentioned. Please, clarify.

4.    A short comment about the correspondence of the traditional use to the modern pharmacological findings is welcome.

5.    Concerning saikosaponins, the class of triterpenoid saponins should be introduced in the Phytochemistry section.

6.    The cytotoxic/antitumor activity and antioxidant properties should be justified with the cell lines/experimental models and signaling pathways, and antioxidant mechanisms, respectively.

 Moderate editing of English language

Author Response

The manuscript is well organized and brings inside into the traditional and modern use, secondary metabolites, and systematic approach for controlling the bolting and flowering of Apiaceae species in TCM.

Some minor concerns arise:

1. As the discussed plants are related to the TCM, the last should be embedded into the title.

Thanks for your suggestion, the title has been revised to “Apiaceae medicinal plants in China: A review of traditional uses, phytochemistry, bolting and flowering, and controlling approaches”. (Page 1, line 2)

2. The term secondary (specialized) metabolites should be embedded into the manuscript (kinds of metabolites should be omitted).

Thanks for your suggestion, “bioactive metabolite” and “kinds of metabolites” have been revised to “metabolite” or “secondary metabolites”. (Page 17, lines 189 and 194; Page 25, line 405; Page 27, line 445)

3. Table 1. Rephrase the table title in more explicating manner. Concerning N 35, 36, 65, 66, 69 etc., “same as Bupleurum” is mentioned. Please, clarify.

Thanks for your suggestion, the title of Table 1 has been revised to “The list of the classification, traditional use, modern pharmacological use, main metabolites of the 228 AMPs”. And the “same as Bupleurum” has been revised to “Used as Bupleurum chinense”. (Page 3, line 90)

4. A short comment about the correspondence of the traditional use to the modern pharmacological findings is welcome.

Thanks for your suggestion, the short comment about the correspondence of the traditional use to the modern pharmacological findings has been added in the manuscripts as “In addition, other modern uses are also enriched such as antitumor, bacteriostat, and analgesia. These modern pharmaceutical properties have been demonstrated to be associated with the bioactive metabolites, and several metabolites have been found to be co-existed in the TCMs”. (Page 16, lines 165 to 167)

5. Concerning saikosaponins, the class of triterpenoid saponins should be introduced in the Phytochemistry section.

Thanks for your suggestion, the class of triterpenoid saponins has been introduced in the Phytochemistry section. (Page 20, lines 310 to 313)

6. The cytotoxic/antitumor activity and antioxidant properties should be justified with the cell lines/experimental models and signaling pathways, and antioxidant mechanisms, respectively.

Thanks for your suggestion, the specific models, signaling pathways, and antioxidant mechanisms have been appended in the manuscripts about cytotoxic/antitumor activity and antioxidant properties. (Page 16, lines 171 to 185)

Reviewer 4 Report

The authors summarized the potential application and improve the yield and quality of Apiaceae medicinal plants, the traditional use, phytochemistry, modern pharmacological use, effect of bolting and flowering, and approaches for controlling the bolting and flowering were summarized. It is a well-prepared manuscript that can be published after minor revisions. The comment relates only to the extension of the section relating to the yiels. The authors should focus even more on suitable chemotypes and varieties of Apiaceae plants. Some varieties (e.g. see DOI: 10.1016/j.indcrop.2015.11.090) can provide significantly more biomass, it is also important to mention the possibility of eliciting active substances.

Author Response

The authors summarized the potential application and improve the yield and quality of Apiaceae medicinal plants, the traditional use, phytochemistry, modern pharmacological use, effect of bolting and flowering, and approaches for controlling the bolting and flowering were summarized. It is a well-prepared manuscript that can be published after minor revisions. The comment relates only to the extension of the section relating to the yields. The authors should focus even more on suitable chemotypes and varieties of Apiaceae plants. Some varieties (e.g. see DOI: 10.1016/j.indcrop.2015.11.090) can provide significantly more biomass, it is also important to mention the possibility of eliciting active substances.

Thanks for your suggestion, the reference (DOI: 10.1016/j.indcrop.2015.11.090) with the related modern pharmacological use  and main metabolites has been cited in the manuscripts in Table 1. (Page 8, No.112)

Round 2

Reviewer 1 Report

Manuscript is significantly improved by the authors and now can be accepted in its current form.

Minor edits required